# MXene: Evolutions in Chemical Synthesis and Recent Advances in Applications

**Sayani Biswas and Prashant S. Alegaonkar ***

Department of Physics, Central University of Punjab, Ghudda 151401, India; sayanibiswas8820@gmail.com
* Correspondence: prashant.alegaonkar@cup.edu.in; Tel.: +91-8830411140

**Abstract:** Two-dimensional materials have secured a novel area of research in material science after the emergence of graphene. Now, a new family of 2D material-MXene is gradually growing and making itsmark in this field of study. MXenes since 2011 have been synthesized and experimented on in several ways.The HF treatment although successful poses some serious problems that gradually propelled the ideas of new synthesis methods. This review of the literature covers the major breakthroughs of MXene from the year of its discovery to recent endeavors, highlighting how the synthesis mechanisms have been developed over the years and also the importance of good characterization of data. Results and properties of this class of materials arealso briefly discussed alongwith recent advance in applications.

**Keywords:** etching; intercalation; delamination; thin films; 2D material; DFT calculations

## 1. Introduction

The interesting and unique physical properties of mono-layered graphene since its discovery in the year 2004 (Novoselov et al., 2004) [1] have given the family of two-dimensional (2D) materials a reputable place both in academics and in the market. Inquisitive minds boosted by the drive to study other dimensions and with sound scientific backup led the new wave of research on previously known 2D materials like hexagonal boron nitride (h-BN), metal oxides, hydroxides, and metal dichalcogenides, as well as propelling many more new recipes for 2D materials. The field being an emerging one, most materials remain in the genre of pure academic interest. Howeverfew of them have shown promising characteristics that put them in the limelight as candidates for commercially viable practical applications. Among these chosen ones are ternary carbides and nitrides/carbonitrides of transition metals named as MXenes (pronounced "maxenes"), the new addition to the constellation 2D material.

Two dimensional solids are crystals with their lateral dimensions considerably larger compared to their thickness, or it can be said that they possess a higher aspect ratio. Their thickness corresponds to just one atomic layer, which makes them distinctive not only in appearance, but also in properties from their bulk form. The transition from bulky 3D materials to 2D ones substituting them is because of the emerging nanotechnology and miniaturization concept. Devices are scaled down dimensionally to ease portability and ease of using without compromising the actual performance. It has been already discovered that graphene, the most studied 2D material, comprising of $sp^2$ bonded carbon atoms connected via aromatic in plane bonds shows excellent electronic properties, [1] (Novoselov et al., 2004), [2] (Geim & Novoselov, 2010), which concluded that graphene might be the best possible metal for metallic transistor applications. Not only had it had an upper hand on size scalability, but also offered high sustainable currents more than $10^8$ A/cm$^2$, ballistic transport, and linear I-V characteristics. This study is now being expanded and aimed at other 2D materials, which have shown promising results, demanding more attention towards this genre of material research. Lately, a new class of 2D material—MXene—is gaining reputation and attention with much enhanced and

unique features. In 2011, a research team led by Professor Yury Gogotsi at Drexel University in Philadelphia, USA, discovered MXenes while working on selective extraction of metal from metal carbides to make CDC (carbide-derived carbon produced by the thermal or thermo-chemical removal of metals from carbides) [3] (Lukatskaya et al., 2014). Since then, MXenes have become a fast-growing 2D materials family. They are obtained by selective etching of A layers from 3D layered solids known as the MAX phases, which have a combination of metallic, covalent, and ionic bonds and are metallically conductive [4] (Radovic & Barsoum, 2013). They are labeled after their composition $M_{n+1}AX_n$, where M is an early transition metal (left side of the periodic table from group 3 to group 7), A is a group 13 or 14 element (IIIA or IVA) on the periodic table and X is carbon or nitrogen with n = 1, 2, 3 generally as shown in Figure 1. Structurally, the early transition metals like Ti, Sc, Mo, etc. (indicated by deep red boxes in Figure 1) are fixed to carbon and nitrogen with an A element like Al or Si typically (indicated by blue boxes in Figure 1). The M-X bond nature is strong and directional covalent and the M-A bond is weaker than the M-X bonds [4] (Radovic & Barsoum, 2013). MAX phases come in a wide variety in terms of combinations of M, A, and X atoms. The different bond strengths in MAX phases make them the perfect candidates for top-down approaches applied to obtain 2D materials. They perfectly bridge the gap between metal and ceramics. The glaring difference and arguably the best one that sets MAX phases apart from other ceramics orhigh-temperature alloys is the ease with which they can be mechanically handled. The machined down MAX phases undergoing mechanical deformation can also lead to delamination partially producing lamellas of thickness in the nanometer range. However, the selective removal of 'A' layer was never experimented on to produce 2D nanosheets resembling graphene. Removing both M and A elements from MAX phases leads to the formation of porous CDC. Till now, all the known MAX phases have a hexagonal-layered $P6_3$ symmetry (Figure 2) with X atoms filling the octahedral sites and closed packed M layers. Almost 75 MAX phases are listed until now both in bulk and thin film form, but keeping in mind that synthesis with different combinations and ratios of M, A, and X atoms are possible, this family is expected to have more diversity. Recently, there have been several theoretical predictions made on double transition metal MAX phases and several attempts made to materialize blends like $(Ti_{0.5}Nb_{0.5})_2AlC$ [5] (L. Zheng et al., 2010), $Mo_2ScAlC_2$ [6] (Meshkian, Tao, Dahlqvist, Lu, Hultman, & Rosen, 2017), $(V_{0.5}Cr_{0.5})_3 AlC_2$ [7] (Y. Zhou, Meng, & Zhang, 2008), and many more. MAX phases are mechanically robust and resistant to normal shear [8] (Barsoum & Radovic, 2011) compared to other studied low-dimensional materials like graphene, in which the bond is Vander Waal's in nature.However, MAX phases are more experimentally malleable because of their structure and the presence of different kinds of bond with varying degree of strengths. This feature makes them perfect candidates for selective etching of 'A' layers by chemical etchants subjected to favorable conditions without causing any major change in the M-X bonds. This is the basic foundation of MXene synthesis.

The general formula of MXenes turns out to be $M_{n+1}X_nT_x$, where $T_x$ denotes the surface termination groups like –O, –OH, –F, or –Cl depending on the environment and chemicals used [9] (Benchakar et al., 2020). The major hindrance in the embodiment of the ways applied is to predict and test the stability and reliability of these materials. The recipe needs to be precise as well as simple with no room for mistakes. Selection of etchants and having the right balance of temperature is crucial to selectively etch the A layers failing, which can lead to undesirable results. As opposed to most other 2D materials, including graphene, MXenes proved to be versatile, possessing unique combinations like high electrical conductivity (~6000–8000 S/cm) and hydrophilicity of surfaces, robustness, ability to absorb electromagnetic waves, and high negative zeta potential allowing the formation of stable colloidal solutions in water.The hydrophilicity is a direct contrast to graphene due to the surface termination –O or –OH acquired during synthesis. The negative zeta potential of MXenes is also attributed to the surface groups –Cl, –F, –O, and –OH. When MXenes are synthesized, there is usually a combination of different surface

groups that makes their surface highly negative. This inclines MXenes towards mixing methods incorporating solution base [10] (Wyatt, Nemani, & Anasori, 2021).

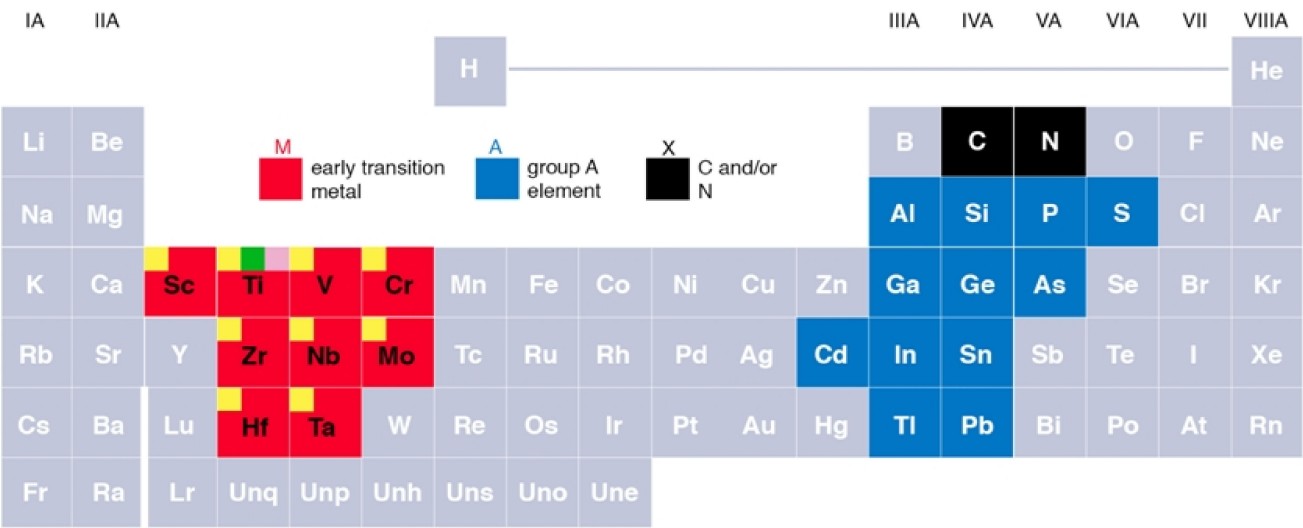

**Figure 1.** Periodic table depicting the elements required in MAX phase formation. The red boxes denote early transition metal (M), blue boxes denotes group 13 and 14 elements (A), and black boxes denotes carbon or nitrogen (X). Reprinted with permission from ref. [4] (Radovic & Barsoum, 2013), Copyright Radovic; TAMU.American Ceramic Society.

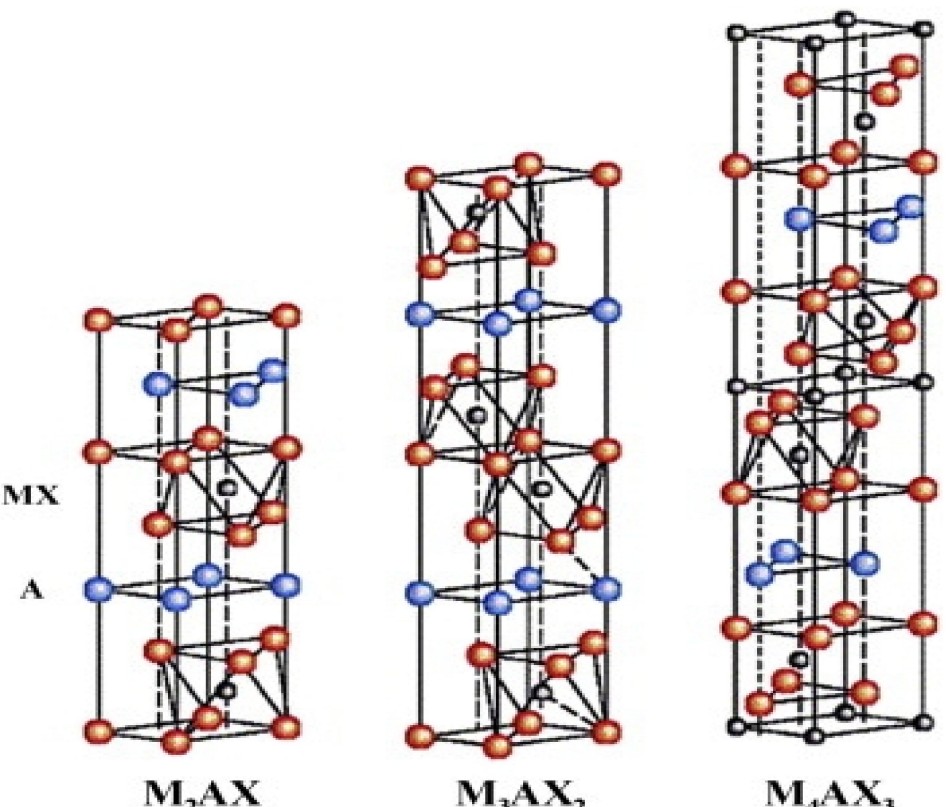

**Figure 2.** MAX phase P63/mmc symmetry. Blue balls denote the A layer atoms, red balls denote the M layer atoms, and gray balls occupying the octahedral site denote the X atoms. Reprinted with permission from ref. [11] (Eklund, Beckers, Jansson, Högberg, & Hultman, 2010). Copyright Elsevier Science B.V.

MAX phases are not HER (hydrogen evolution reaction) because of their 3D structure and lack of exposed surface. On the other hand, MXenes possess activity not only in HER reactions, but are also theoretically predicted to have OER, ORR, nitrogen fixation, $C_2RR$ (carbon dioxide reduction reaction) activity. These bestow the catalytic behavior to MXenes and thus make it valuable in energy storage mechanisms.

Good flexibility owing to their 2D morphology and sheet-like structure is the particular feature of MXenes that makes them outstanding for composite formation. Because of this morphology MXenes provide a larger surface area or exposed surface for reactions to carry on or for bond formation with other polymers. Also it has been established that the surface terminations are the root cause of hydrophilicity, which in turn makes bond formation easier. While extraction of monolayer of a MXene can be tricky, if obtained can lead to more applications in the nanomaterial industry. The trend of miniaturization needs suitable materials that can be molded or at least scaled down in size easily without changing the overall performance. These ideas provide opportunities of blending excellent properties of different materials constructively. Although being in the early stage of research, MXenes are already considered as replacements for other conventional 2D materials. For example, in the case of graphene we see that the conductivity is high, but the bare surface is hydrophobic, making it incapable of sustainable bond formation. Mxenehas the unique combination of both high electrical conductivity as well as hydrophilicity. In a study, it wastheoretically shown that $Ti_3C_2$ Mxenenanosheetsare superior hybridization matrix than graphene, which makes MXene nanosheets and composites better adsorbents. This is due to the weak layer stacking, hydrophilic nature, and $\pi$ electron free rigid free nanosheets [12] (Jin et al., 2018). Like their precursors, MXenes are tunable too in terms of composition, structure and properties. The interlayer distancewhich can be manipulated plays a huge role in catalysis and energy storing. The nature of surface terminations dictates the nature of bond formation and also charge transport in them. So, we get an idea that by making necessary changes we can obtain results as per our requirements.MXenes have varied applications with promising performances in: (i) Energy sector as energy storage devices like supercapacitors [13] (Lukatskaya et al., 2013), Li-sulphur batteries [14] (X. Liang, Garsuch, & Nazar, 2015); (ii) Environmental related application like water desalination [15] (Ren et al., 2015); (iii) EMI shielding and printable antennas [16] (Shahzad et al., 2016); (iv) Catalysis [17] (Seh et al., 2016); (v) Biomedical [18] (Anasori & Gogotsi, 2019); in addition to numerous other applications [19] (Anasori, Lukatskaya, & Gogotsi, 2017). In the following sections, an overview of different synthesis methods developed over the years has been discussed along with the significance of characterization and brief highlights of properties. Applications are also presented alongwith recent researches that arebeing carried out.

## 2. Synthesis of Mxene
### 2.1. Etching

The separation of a single or more than one atomic layer from compounds stacked as sheets, in which the bonding between the layers is weaker compared to the bonds within the layers, is a basic approach for obtaining 2D materials. The other materials of this class before the existence of MXenes were mostly derived from layered solids with hydrogen or Vander Waals interlayer bonds like hexagonal boron nitride [20] (Satawara, Shaikh, Gupta, & Gajjar, 2020) and molybdenum disulfide ($MoS_2$) [21] (Zuoli He & Que, 2016). MXene synthesis is attainable via wet chemical etching of selective layers from their precursor material MAX phase with formula $M_{n+1}AX_n$ at room temperature. MAX phases comprise of stronger layer-to-layer bonds in comparison to others demanding stronger etchants. The 'A' layers (in most cases Al) are corroded under specific synthesis conditions. The resulting $M_{n+1}X_n$ layers are stable chemically compared to the 'A' layer of atoms. The first and the most used chemical etchant is HF (hydrofluoric acid) and was used on the MAX phase $Ti_3AlC_2$ to obtain $Ti_3C_2$Mxene [22] (Naguib et al., 2011a). A schematic diagram of Mxene formation using HF is depicted in Figure 3 for better understanding the process. Soon after the concentration of HF was changed to obtain further unique combinations

and throughout the years, many other methods have been employed, for which brief representations of these are given in Table 1. By varying M and X species, the surface chemistries contributing to different functional groups and also the number of layers 'n' in $M_{n+1}X_n$, tuning of MXene properties and applicability is possible. Initially, the MAX phases are soaked for a certain time and with optimum agitation in specific acids that destroys the M-A bonds. There are many guidelines and protocols in the literature regarding MXene production [9] (Benchakar et al., 2020). In order to fix the exact synthesis conditions, there should be a clear picture of what is desired as a result of the experiment and to be precise about the performance in their application. The synthesis process can run for few hours or it can take few days depending on the factors: concentration of etchants and etching time.

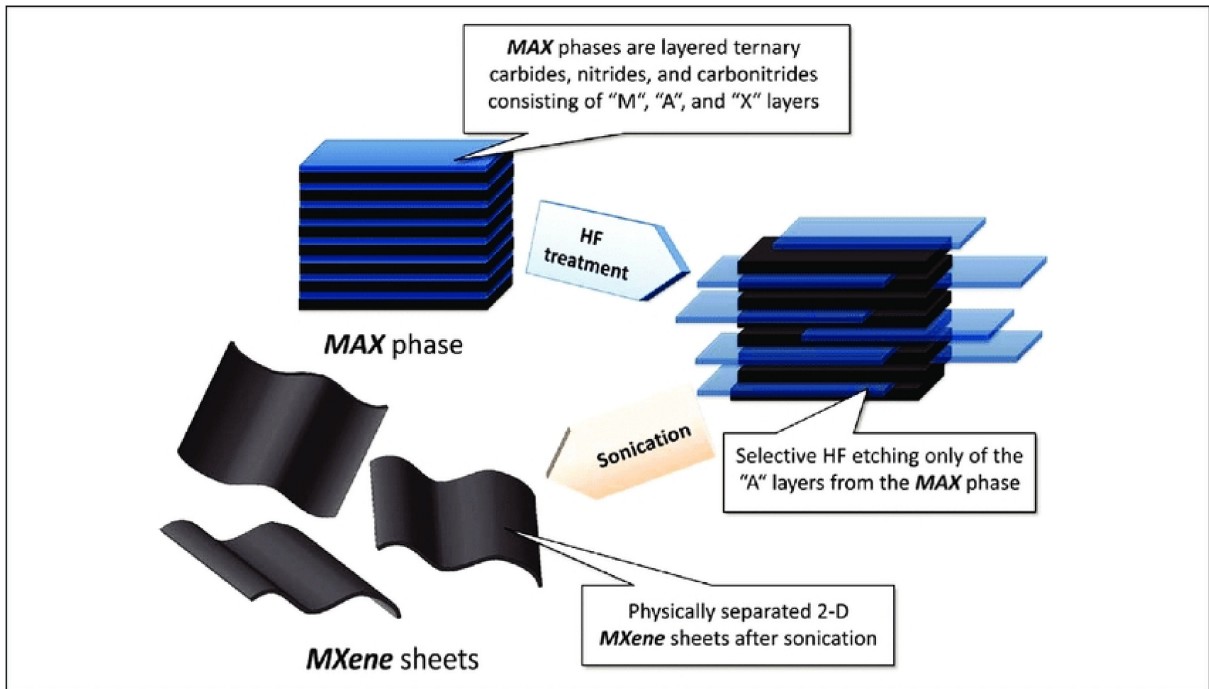

**Figure 3.** Schematic diagram of MXene formation. The blue-black layered structure (MAX phase) is treated with HF, which selectively removes the blue layers (A layer etching). The black multi-layered structure is then sonicated to yield 2D MXene sheets. Reprinted with permission from ref. [23] (Naguib, Mochalin, Barsoum, & Gogotsi, 2014). Copyright 2013 Wiley-VCH Veralg GmbH &Co. KGaA.

Two-dimensional nanocrystals were produced by treating $Ti_3AlC_2$ [24] (X. Wang & Zhou, 2010) powders with HF (50% concentrated) at room temperature for almost 2 h by Naguib et al. [22] (Naguib et al., 2011a). The immersion of $Ti_3AlC_2$ in HF can be summarized in the following simplified reactions.

$$Ti_3AlC_2 + 3HF = AlF_3 + Ti_3C_2 + 3/2 \, H_2 \tag{1}$$

$$Ti_3C_2 + 2H_2O = Ti_3C_2(OH)_2 + H_2 \tag{2}$$

$$Ti_3C_2 + 2HF = Ti_3C_2F_2 + H_2 \tag{3}$$

The first Reaction (1) is the mandatory one followed by Reactions (2) and (3) and the result is 2D $Ti_3C_2$ layers with F and/or OH surface functional groups. In succeeding time, HF etching method was successfully employed on other MAX phases that gave us a family of new MXenes such as $Nb_2CT_x$ [25] (Naguib et al., 2013), $Ti_2CT_x$ [26] (K. Zhu et al., 2019), $V_2CT_x$ [25,27] (Naguib et al., 2013; VahidMohammadi, Hadjikhani, Shahbazmohamadi, & Beidaghi, 2017), $(V_{0.5}Cr_{0.5})_3C_2T_x$ [28] (Naguib et al., 2012), and many more as summarized in Table 2 with proper working conditions and references. As stated earlier; particle size,

time of etching, and HF concentrations are the parameters of successful conversion of MAX Phase to Mxene [29] (Munir et al., 2020).

**Table 1.** Timeline of evolution of MXene and important breakthroughs in this field.

| Year | Synthesis Methods Development Timeline |
|---|---|
| 2011 | Discovery of $Ti_3C_2$MXene via HF etching method. |
| 2012 | Emergence of MXene family including $Ti_2C$, $Ta_4C_3$, $Ti_3CN$, $(V_{0.5}Cr_{0.5})_3C_2$, $(Ti_{0.5}Nb_{0.5})_2C$. |
| 2013 | Delamination of MXenes. |
| 2014 | (i) Bifluoride $NH_4HF_2$etching method; (ii) LiF/HCl etching method producing clay (small flakes); (iii) MXene-polymer composite. |
| 2015 | (i) TBAOH and amine-assisted delamination; (ii) Urea Glass Route to produce $MO_2C$ and $MO_2N$ Mxene; (iii) Chemical vapor deposition method, theoretical predictions of double-ordered MXenes. |
| 2016 | (i) Molten salt etching method to produce nitride-based MXene $Ti_4N_3$; (ii) Room temperature synthesis of new MXene $Zr_3C_2$. |
| 2018 | (i) Hydrothermal synthesis method using NaOH; (ii) Electrochemical fluoride free synthesis of MXene using a binary aqueous system. |
| 2019 | (i) Advances in MXene-composite synthesis; (ii) Lewis acid molten salts etching method to synthesize Mxenes |
| 2020 | Synthesis of MXene/PANI composite. |

**Table 2.** Summary of MXenes prepared by HF treatment. Etchant used—HF (10% to 51%). Temperature—RT.

| S. No. | Precursors | Mxene | Conditions | | |
|---|---|---|---|---|---|
| | | | Time (h) | Yield (%) | Ref. |
| 1. | $Ti_2AlC$ | $Ti_2CT_x$ | 10 | 80 | [26] |
| 2. | $V_2AlC$ | $V_2CT_x$ | 90 | 60 | [27] |
| 3. | $Nb_2AlC$ | $Nb_2CT_x$ | 90 | 100 | [25] |
| 4. | $Ti_2AlN$ | $Ti_2NT_x$ | 24 | NA | [30] |
| 5. | $Mo_2Ga_2C$ | $Mo_2CT_x$ | 3 | NA | |
| 6. | $(Ti_{0.5}Nb_{0.5})_2AlC$ | $(Ti_{0.5}Nb_{0.5})_2CT_x$ | 28 | 80 | [28] |
| 7. | $Ti_3AlC_2$ | $Ti_3C_2T_x$ | 2 | 100 | [22] |
| 8. | $(V_{0.5}Cr_{0.5})_3AlC_2$ | $(V_{0.5}Cr_{0.5})_3C_2T_x$ | 69 | NA | [28] |
| 9. | $Ta_4AlC_3$ | $Ta_4C_3T_x$ | 72 | 90 | [28] |
| 10. | $Nb_4AlC_3$ | $Nb_4C_3T_x$ | 96 | 77 | [31] |
| 11. | $V_4AlC_3$ | $V_4C_3T_x$ | 165 | NA | [32] |
| 12. | $Ti_3AlCN$ | $Ti_3CNT_x$ | 18 | 80 | [28] |
| 13. | $Mo_2TiAlC_2$ | $Mo_2TiC_2T_x$ | 48 | 100 | [33] |
| 14. | $Mo_2Ti_2AlC_3$ | $Mo_2Ti_2C_3T_x$ | 90 | 100 | [33] |
| 15. | $(Mo_{2/3}Y_{1/3})_2AlC$ | $Mo_{4/3}CT_x$ | 6072 | NA | [34] |
| 16. | $(Nb_{2/3}Sc_{1/3})_2AlC$ | $Nb_{4/3}CT_x$ | 30 | NA | [35] |
| 17. | $(W_{2/3}Sc_{1/3})_2AlC$ | $W_{4/3}CT_x$ | 30 | NA | [36] |
| 18. | $Zr_3Al_3C_5$ | $Zr_3C_2T_x$ | 60 | NA | [37] |
| 19. | $Hf_3[Al(Si)]_4C_6$ | $Hf_3C_2T_x$ | 60 | NA | [38] |

For industrial purposes, however, HF treatment can be a challenge. If large scale production of MXene is to be considered, then safer routes must be sought out. We are edging towards the norms where environmental safety shall come first when planning out any form of industrial endeavors. Using HF can be hazardous to people as well as the environment as it can leave toxic waste behind to dispose of. This is why it has been

the first priority and concern for every researcher studying MXenes to replace HF as the etching agent.

In Table 2, we clearly see that although all of the precursors have Al in common as the etched out layer, the etching conditions vary widely, which indicates the difference in M-Al bond energies in different MAX phases. Also, the variable 'n' in $M_{n+1}AlC_n$ phase dictates a great deal about the conditions to be applied. The higher the value of 'n' the more stable the MXene is and so needs a greater concentration of HF. This information is very crucial for further studies in synthesis. We have to choose the precursor wisely to get the desirable MXene, keeping in mind the economic feasibility when produced at the industrial level. This also sheds light to the already explored but neglected group of MAX phases. It is equally important to study MAX phases in order to understand and apply MXenes in various ways.

2.1.1. Modification of Acid Etching Methods

In 2014, a trend of finding a replacement for HF etchant spurred the study of some different synthesis techniques because HF is harmful to the body, which makes handling the chemical a tough challenge and also can produce toxic wastes that effect the environment. Attempts were made to search for milder etchants with minimum waste and less toxicity. Halim et al. [39] (Halim et al., 2014) showed that Ammonium Bifluoride ($NH_4HF_2$) to be a good candidate. In the report, at first, epitaxial $Ti_3AlC_2$ thin films were deposited on sapphire substrates at 780 °C. Then, a comparative study was carried out, the first one by using traditional 50% concentrated HF on samples of thickness 15, 28, 43, and 60 nm for 10, 15, 60, and 160 min, respectively. A total of 1 M of $NH_4HF_2$ was used in the second study, with the sample thicknesses mentioned above but for 150, 160, 420, and 660 min, respectively. This method is a one-step synthesis method for MXenes as there was simultaneous intercalation of ammonia (cations) during the etching process itself, which is a great advantage over HF etching.

The operative reactions are:

$$Ti_3AlC_2 + 3NH_4HF_2 = (NH_4)_3AlF_6 + Ti_3C_2 + 3/2\ H_2 \tag{4}$$

$$Ti_3C_2 + aNH_4HF_2 + BH_2O = (NH_3)_c(NH_4)_d Ti_3C_2(OH)_xF_y \tag{5}$$

Reaction (5) shows that $NH_3$ and $NH_4^{+1}$ get intercalated between $Ti_3C_2T_x$ layers, which can alter MXene properties significantly. This intercalation swells the MXene as the distance between the layers increases (the c lattice parameter was increased by 25% than the films etched using HF). $NH_4HF_2$ is less hazardous than HF and is therefore a milder etchant. In 2017, Feng et al. further reported that the surface functional groups present on flakes makes the $Ti_3C_2T_x$ surface negative, as a result the cations ($NH_4^+$) are attracted to the negative surface and gets attached onto the surface, enlarging the c lattice parameter of $Ti_3C_2T_x$ accordingly [40] (Feng et al., 2017). The paper also reported the optimal reaction conditions of $NH_4HF_2$ exfoliation: temperature 60 °C, etching time 8 h, and average diameter of $Ti_3AlC_2$ to be 325 mesh.

In the same year, seeking for a safer route, Ghidiu et al. made use of abundant and commercially feasible HCl and fluoride salts for synthesis of two-dimensional titanium carbide with high volumetric capacitance [41] (Ghidiu, Lukatskaya, Zhao, Gogotsi, & Barsoum, 2014). This is also a one-step synthesis method with exotic properties. LiF was dissolved in 6M HCl and then $Ti_3AlC_2$ powder was gradually added to the solution. The mixture was subsequently heat treated at 40 °C for 45 h. After etching and washing the sediments several times, clay-like paste was obtained, which could be rolled and molded to any shape when hydrated as shown in Figure 4. This was the first instance brought to attention about the analogous nature of MXenes and clays. The intercalation of water in this case acts as the lubricant that allows facile shearing, which reduced the typical sonication time for delamination from order of 4 h to 30–60 min. LiF can be replaced by other fluoride salts such as NaF, KF,CsF and $CaF_2$ in HCl or $H_2SO_4$ [41] (Ghidiu et al., 2014). Although

both methods are safer and easier to perform, they were still accompanied by the in situ formation of HF. In 2017, MILD (minimally intensive layer delamination) technique was optimized using (12M LiF/9M HCl) at room temperature [42] (Alhabeb et al., 2017), which provided single large flakes with less structural defects. These methods are also known as "in situ HF method", in which HF can be replaced by the combination mixture of fluoride salts (LiF, NaF, KF, and $FeF_3$) and HCl [43] (X. Wang et al., 2017).

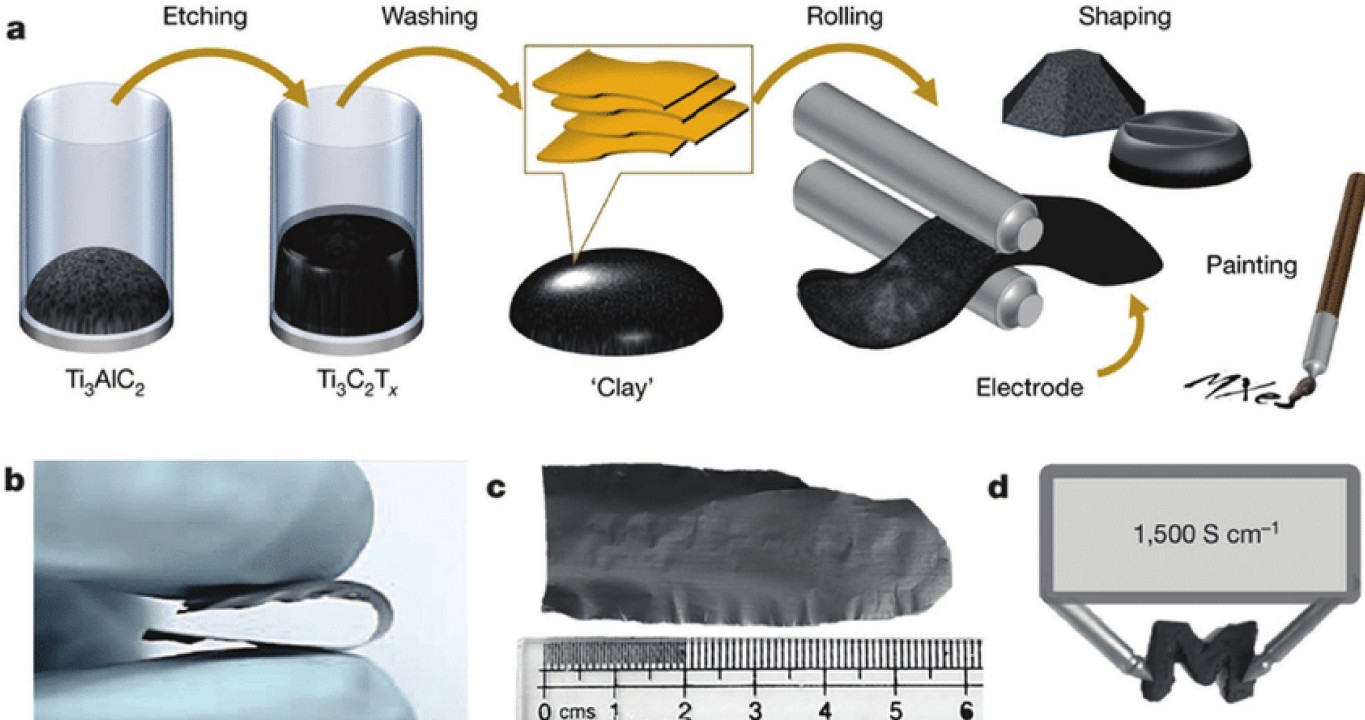

**Figure 4.** Free-standing titanium carbide clay film. (**a**) Schematic diagram of MXene formation with MAX phase powder being etched and then subsequently washed to remove impurities. The resultant product behaves like clay, which can be rolled and is flexible enough to make free-standing films. (**b**) If the dried samples are hydrated; swelling happens immediately. Again drying makes them shrink (**c**) Rolled film image. (**d**) The clay is bent to give it an 'M' shape. Then it is dried forming a conductive solid. Reproduced with permission from [41] (Ghidiu et al., 2014). Copyright 2014, Nature Publishing Group.

### 2.1.2. Fluoride-Free Synthesis Methods

Three prime synthesis pathways have been constantly explored to avoid the use of HF:Hydrothermal synthesis; synthesis via CVD process and electrochemical synthesis at room temperature in HCl or ammonium chloride/tetramethylammonium hydroxide (TMAOH) electrolytes.

Xu et al. in 2015 developed a versatile chemical vapor deposition (CVD) process with a Cu foil mounted on Mo foil serving as the substrate and methane as the source of carbon at temperatures above 1085 °C [44] (C. Xu et al., 2015). The aim was to grow supreme quality 2D ultrathin alpha-$Mo_2C$ crystals a few nanometers thick. Tuning the experimental conditions can create variations in size and thickness of the films. The nucleation density was directly proportional to growth temperature and the lateral size increases with growth time. However, this method gave a very low yield and consumes high energy, which restricts its further development. The synthesis of $Mo_2C$ via CVD process had some set-backs as during the process catalysts are produced that have low crystallinity, heterogeneous structures, and are characterized by varied sizes and compositions (polydispersive in nature). Also, the deactivation of active catalytic sites due to the production of amorphous carbon poses

as a hindrance. Later that year, Ma et al. proposed a study that took care of these limitations and prepared $Mo_2C$ and $Mo_2N$ nanomaterial using a simple "urea glass" route [45] (Ma, Ting, Molinari, Giordano, & Yeo, 2015). Precursor element chosen was 1.45 M solid metal $MoCl_5$ and ethanol was added to it. The reaction yielded Mo-orthoesters. Solid Urea in varying ratios was then added to the alcoholic solution and the mixture was stirred till the complete solubilization of urea (takes usually 1 h). Here, urea played the double role: (1) as the C/N source (2) as the balancing agent during the necessary thermal exposure of the precursor mixture to convert it into metal carbide or nitride with zero or negligible amount of residual C atom. The precursors had gel-like texture and were taken in different crucibles which were further heated at 800 °C for 3 h under $N_2$ gas flow. The end product was silvery black powder of alpha-$Mo_2C$ and gamma-$Mo_2N$, which after a series of characterizations are presented as the most efficient HER catalysts.

Theoretical predictions of 2D double-ordered MXenes also came into limelight in 2015. Predicting the existence of two new branches of 2D-ordered carbides-$M'_2M''C_2$ and $M'_2M''_2C_3$, where $M'$ and $M''$ are two different early transition metals was made using density functional theory (DFT) [33] (Anasori et al., 2015a). Validation through DFT predictions of $Mo_2TiC_2T_x$, $Mo_2Ti_2C_3T_x$, and $Cr_2TiC_2T_x$ were reported (where T is the surface termination). The chemical and electrochemical properties of the 2D flakes were governed by the Mo and Cr atoms present on the outside layer. The stability of over 20 new ordered MXenes were proved. According to the study, the following MXenes are categorized as:

Stable in fully ordered state at 0K: $Mo_2TiC_2$, $Mo_2VC_2$, $MO_2TaC_2$, $Mo_2NbC_2$, $Cr_2TiC_2$, $CrVC_2$, $Cr_2TaC_2$, $Cr_2NbC_2$, $Ti_2NbC_2$, $Ti_2TaC_2$, $V_2TaC_2$, and $V_2TiC_2$.

Stable in partially ordered state at 0K: $Nb_2VC_2$, $Ta_2TiC_2$, $Ta_2VC_2$, and $Nb_2TiC_2$.

Later in 2016, theoretical prediction and discussion of materials synthesis of a chemically ordered MAX phase, $Mo_2ScAlC_2$, and its two-dimensional derivate $Mo_2ScC_2$ MXene was put forward based on the ab-initio calculations in DFT [6] (Meshkian, Tao, Dahlqvist, Lu, Hultman, & Rosén, 2017).

MAX phases with the A-site occupied by a late transition metal like Fe, Pt, Zn, and Ni was a rarity and difficult to synthesize, but they are expected to manifest multiple functional properties such as catalysis and magnetism. Attempts were made by Fashandi et al. to obtain MAX phases of noble-metal elements using a replacement reaction [46,47] (Fashandi, Dahlqvist, et al., 2017; Fashandi, Lai, et al., 2017). Si was replaced by Au in the A-site of $Ti_3SiC_2$ at high annealing temperature by a thermodynamic driving force for separating Au and Si. In similar context, Wang et al. also successfully applied the replacement theory between $SnO_2$ and Al in $Ti_3AlC_2$ [48] (S. Wang et al., 2017). These A-site exchange methods of producing novel MAX phases can be extended as an approach to prepare MXenes by A-site group etching process. Based on this idea, Li and colleagues synthesized a series of novel nanolaminated MAX phases and their corresponding MXenes; by exchanging elements from the A layer in traditional available MAX phases. $ZnCl_2$, an example of late transition metal halide was used in this study due to their behavior as so called Lewis acid in their molten state [49] (M. Li et al., 2019). They act as electron accepting ligands in their molten state and underwent a thermodynamical reaction with the A element in the MAX phase. Traditional MAX phase precursors ($Ti_3AlC_2$, $Ti_2AlC$, $Ti_2AlN$, and $V_2AlC$—labeled as Al-MAX) were made to react with $ZnCl_2$ (the molar mixture of Al-MAX $ZnCl_2$ = 1:1.5). For synthesizing the MXenes with -Cl terminations, the mixture powder comprised of Al-MAX:$ZnCl_2$ = 1:6 as a starting material. The mixture was heat treated at 550 °C for different time ranging from 0.5 to 5 h. $Ti_3C_2Cl_2$ and $Ti_2CCl_2$MXenes were obtained by further exfoliating the novel MAX phases $Ti_3ZnC_2$ and $Ti_2ZnC$ in molten $ZnCl_2$. This was the first time that Cl-terminated MXenes were achieved via a fluoride-free synthesis method. This new way can pave the path for better applicability of MXenes in energy storage and enhanced electrochemical features.

Another non-fluorine electrochemical approach was coined by Yang et al. based on thecorrosion of $Ti_3AlC_2$posing as anode in binary aqueous electrolyte [50] (S. Yang

et al., 2018). In atwo-electrode system, two pieces of bulk $Ti_3AlC_2$ were fitted as both the anode (which wascorroded by the etching reactions) and the cathode (placed as a counter electrode). The choiceof electrolyte was the difficult task. Etching effects of chloride containing electrolytes wasconsidered because Cl-ions can bind strongly with Al [51] (Toušek, 1980). Since etching reactionpreferentially occurs on the surfaces, the idea of intercalating the etched $Ti_3C_2T_x$ layers was put forward based on the theory that $Ti_3C_2T_x$ flakes can easily uptake cations like $NH_4^+$. So, the tailoring of the electrolyte was done by selecting 1M $NH_4Cl$ and 0.2 M tetramethylammonium hydroxide (TMAOH) keeping the pH value greater than 9. Potential of +5 V was applied followed by the detachment of the bulk anode turning the transparent electrolyte ash grey with huge amount of gelatinous precipitate suspended in it. Delamination of the grinded $Ti_3C_2T_x$ powder was done by adding it in 25 wt% TMA-OH to get individual 2D sheets. This was a very effective alternative method with a high yield (>90%).

Hydrothermal process is another appealing strategy to fabricate MXenes. Wang et al. introduced a simple and new one-pot synthesis process of producing $Ti_3C_2T_x$ MXene with $NH_4F$ as the etching agent in a hydrothermal environment [52] (L. Wang et al., 2016). The precursor powder $Ti_3AlC_2$ was pre made. In 60 mL of deionized water, 5 g of $NH_4F$ was dissolved by magnetic stirring and in the middle of vigorous stirring; 0.5 g of $Ti_3AlC_2$ powder was slowly added. Subsequent centrifugation and washing with deionized water and ethanol yielded black precipitate of $Ti_3C_2T_x$ which was then left to dry at 60 °C for 12 h. As a conclusion, it was stated that the growth of the nanomaterial was greatly affected by $NH_4F$ concentration, temperature, and time of reaction. Improved adsorption performance of synthesized $Ti_3C_2$ and $Nb_2C$ 2D MXenes was reported in a paper by another via solvothermal treatment in a mixture of hydrochloric acid and sodium tetrafluoroborate ($NaBF_4$) [53] (C. Peng et al., 2018). The aim of this work was providing a hydrothermal environment to avoid the exposure to toxic HF gas, to develop a universal and novel route to prepare MXenes and to explore the reaction mechanism of hydrothermal etching route. High purity MAX phase powders $Ti_3AlC_2$ and $Nb_2AlC$ were first prepared. Then, a comparative study was run, one by traditional HF etching method (t-MXene) and the other through hydrothermal technique (h-MXene). The MXene flakes were then prepared by sonication-assisted delamination using DMSO. The following reactions were involved:

$$NaBF_4 + HCl \rightarrow HBF_4 + NaCl \tag{6}$$

$$HBF_4 \rightarrow HF + BF_3 \tag{7}$$

$$BF_3 + 3H_2O \rightarrow 3HF + H_3BO_3 \tag{8}$$

$$Ti_3AlC_2 + 3HF \rightarrow AlF_3 + 3/2\ H_2 + Ti_3C_2 \tag{9}$$

$$Ti_3C_2 + 2H_2O \rightarrow Ti_3C_2(OH)_2 + H_2 \tag{10}$$

$$Ti_3C_2 + 2HF \rightarrow Ti_3C_2F_2 + H_2 \tag{11}$$

$$Nb_2AlC + 3HF \rightarrow AlF_3 + 3/2\ H_2 + Nb_2C \tag{12}$$

$$Nb_2C + 2H_2O \rightarrow Nb_2C(OH)_2 + H_2 \tag{13}$$

Both $Ti_3C_2$ and $Nb_2C$ have undergone the same process. The HCl and $NaBF_4$react as in Equation (6) forming $HBF_4$which then readily reduced to HF and $BF_3$. So, we notice that there is an in situ HF formation, which then cleaves the $Ti_3AlC_2$ MAX phase and produces $Ti_3C_2$alongwith the generation of $AlF_3$. So, although HF was not directly used, the slow release of F⁻caused the late, but enough, production of HF thatcarried forward the etching reaction.

Li and colleagues also reported an alkali (NaOH)-assisted hydrothermal treatment to prepare a typical MXene—$Ti_3C_2T_x$ ($T_x$ = –OH, –O) [54] (T. Li et al., 2018). This method was inspired by the industrial Bayer process of refining bauxite to produce alumina [55] (Adamson, Bloore, & Carr, 2016; Pearson, 1955). The selective etching of Al from $Ti_3AlC_2$

was carried on successfully in a solution of 27.5 M NaOH at 270 °C that resulted in –OH and –O terminal 92 wt% pure multilayer $Ti_3C_2T_x$ MXene.

### 2.1.3. Molten Fluoride Salt ETCHING Method

Up until 2015, the carbide-based MXenes were the only ones successfully produced by HF etching or other etching methods. Ternary nitride-based 2D MXenes were a rarity and all attempts of synthesis by using HF as the etchant were unfruitful. Naguib et al. in their study revealed that in $Ti_{n+1}AlN_n$ MAX phases, the Al atoms are strongly bonded as compared to $Ti_{n+1}AlC_n$. There are two reasons behind this:

(a) Cohesive energy of $Ti_{n+1}N_n$ < Cohesive energy of $Ti_{n+1}C_n$; which implies a lower stability of structure owing to this the layers dissolve in HF solution.

(b) The formation energies of $Ti_{n+1}N_n$ from $Ti_{n+1}AlN_n$ > the formation of $Ti_{n+1}C_n$ from $Ti_{n+1}AlC_n$, which implies requirement of more energy for extraction of Al atoms from their MAX phases.

Ruling out the failure of HF etching, Urbankowski et al. synthesized $Ti_4N_3T_2$ (2D titanium nitride) via molten salt method. First, the precursor $TiAlN_3$ MAX phase was produced using $TiH_2$, AlN, and TiN powders and mixing them with a molar ratio of 2:1:2 [56] (Urbankowski et al., 2016). Top-down approach of mechanical milling was employed on the mixture for 14 h with subsequent hot pressing for 24 h at 1275 °C and 70 MPa. The fluoride salt mixture contained KF, LiF, and NaF in the ratio of 0.59:0.29:0.12, respectively. Equal masses of $TiAlN_3$ powder and the fluoride sat mixture were then ball milled for 6 h and further thermally treated at 550 °C for 30 min with heating and cooling rate of 10 °C/min.

### 2.1.4. MXenes Prepared from Other Precursors Than Al-MAX Phases

Zhou et al. For the first time reported the preparation of Zirconium (Zr)-based 2D carbide by selective extraction of Al-C groups from an alternatively layered ternary $Zr_3Al_3C_5$ in 2016 [37] (J. Zhou et al., 2016). This study showed that beyond MAX phases, the new family of layered ternary and quaternary compounds labeled as $MnAl_3C_2$ or $Mn[Al(Si)]_4C_3$ can be used as precursor material for MXene derivation. $ZrC_2T_x$ was produced by immersing parent $Zr_3Al_3C_5$ powder in 50% concentrated HF solution causing the removal of (Al-C)$_x$ units. The reaction goes as:

$$Zr_3Al_3C_5 + HF \rightarrow AlF_3 + CH_4 + Zr_3C_2 \tag{14}$$

$$Zr_3C_2 + H_2O \rightarrow Zr_3C_2(OH)_2 + H_2 \tag{15}$$

$$Zr_3C_2 + HF \rightarrow Zr_3C_2F_2 + H_2 \tag{16}$$

Obtaining $Hf_3C_2T_x$ was different as in $Hf_3Al_3C_5$; the interfacial bonds between Al-C units and Hf-C units are very strong. After doping with Si to get $Hf_3[Al(Si)]_4C_6$, the bond strengths between the units effectively reduced which on further HF treatment led to the synthesis of MXenes. DFT calculations showed that all of the prepared functionalized MXene $Zr_3C_2T$ (T = OH, O, F) manifests metallic properties and particularly $Zr_3C_2O_2$ showed strongest mechanical strength. This was somewhat predictable because all the MAX phases that's how magnetic behavior contain Zr or Mn.

Another instance of non-MAX phase material that can be a precursor to MXene was reported by Meshkian and colleagues [6] (Meshkian et al., 2015). $Mo_2C$ (molybdenum carbide) was derived from gallium-based atomic laminate $Mo_2Ga_2C$. Thin films of $Mo_2Ga_2C$ were deposited using DC magnetron sputtering technique from three elemental targets: Mo, Ga, and C. Immersing the films in 50% concentrated HF (aq.) for 3 h at a temperature of 50 °C caused the selective etching of Ga layers. The sample collected was then rinsed in de-ionized water and ethanol. This was the first reported non-Al containing MAX phase giving the first Mo-based MXene, the new addition to the family of 2D materials.

2.1.5. MXene–Polymer Composites

Recently, fabrication of composites is potentially the best approach to develop versatility and enhanced features in material science. MXenes due to their graphene-like morphology, hydrophilicity, layered structure, and good flexibility can be tuned into multifunctional composites. The high aspect ratio of mono-layer nanosheets (delamination making the surfaces more hydrophilic) is perfect as nanofiller in polymer composites with diverse functionality. When specifically, MXenes form composites with polymers, their excellent mechanical properties, hydrophilic surfaces, and ability to conduct similar to metals can enhance the mechanical and thermal properties of polymers. In situ polymerization and ex situ mixing are the two ways to obtain MXene-polymer composites. In-situ polymerization mechanism includes (i) UV radiation [57] (J. Chen et al., 2015) (ii) mild physical agitation [58] (Boota et al., 2016) (iii) electrodepositing [59] (M. Zhu et al., 2016). Ex situ mixing is achieved by solution mixing and filtration method.

Ling et al. was the first to report to produce $Ti_3C_2T_x$/polymer composites using two polymers: an electrically neutral polyvinyl alcohol (PVA) and a charged polydiallyldimethylammonium chloride (PDDA) [60] (Ling et al., 2014). Vacuum-assisted filtration (VAF) was used to fabricate conductive flexible and free-standing films of (i) $Ti_3C_2T_x$, (ii) $Ti_3C_2T_x$/PDDA, (iii) $Ti_3C_2T_x$/PVA. PDDA solution (5 mL; 2 wt% aq. solution) was added dropwise to 35 mL; 0.34 $mgmL^{-1}$ colloidal solution of MXene followed by magnetic stirring for 24 h. The solution was further centrifuged at 3500 rpm for 1 h and then washed with deionized water and preparing for another hour of centrifugation. For the MXene/PVA composite, the MXene to PVA weight ratios taken up were: 90:10, 80:20, 60:40, and 40:60. Both the MXene colloidal solution (0.3 $mgmL^{-1}$) and the PVA (0.1 wt%) aqueous solution were then mixed and the mixture was sonicated for 15 min in a water bath.

Chen et al. in their strictly experimental study first prepared $V_2AlC$ phase using vanadium, aluminum, and graphite flakes, then pulverized the bulk material to prepare $V_2C$ MXenes and finally added 20 mg $V_2C$ and 2 mL of monomer (DMAEMA) in a tube irradiated by UV lamp for 40 min at room temperature [57] (J. Chen et al., 2015). Magnetic stirring of the reaction mixture was performed under dry $N_2$ at room temperature during the irradiation and then was subsequently centrifuged at 8000 rpm for 10 min in methanol. Sedimentation of polymer grafted $V_2C$ was obtained.

Boota and co-workers reported combining electrochemically active polymers with MXenes by in-situ polymerization technique. Conductive polymer polypyrrole(PPy)($C_4H_4NH$) intercalate between $Ti_3C_2T_x$ layers expanding the space between them to facilitate charge transport [61] (C. Chen et al., 2017). Delaminated colloidal solution of $Ti_3C_2T_x$ suspension was mixed PPy in two different mass to volume ratio 2:1 and 1:1 to obtain$Ti_3C_2T_x$ "paper"— refereed as the pristine $Ti_3C_2T_x$ films. The mixture was stirred for 12 h at RT with 1000 rpm subsequently undergoing vigorous stirring to prevent pyrrole monomer agglomeration. Later in 2018, Cao and co-workers also proposed in situ chemical oxidative polymerization-based facile method that modified $Ti_3C_2T_x$ (Cao, Han, Zheng, & Wang, 2019) [62] with 1,5 naphthalene disulfonic acid (NA) and cetyltrimethylammonium bromide (CTAB). This made the $Ti_3C_2T_x$ and PPy to form a condensed sandwich-like structure and also improved the electron transport and capacitance of $Ti_3C_2T_x$/PPy composites. Fixing the $Ti_3C_2T_x$/PPy mass ratio at 20:80, the NA to pyrrole ratio was modified from 0.2:1, 0.4:1, 0.6:1, to 0.8:1. After stirring the mixture for 6 h, APS was added and the reaction went on for 24 h. Washing and filtering using deionized water, the polymer composite NA-$Ti_3C_2T_x$/PPy was finally obtained by drying the sediments for 70 °C, 24 h. The NA-CTAB-$Ti_3C_2T_x$/PPy composite was obtained following the same process only with an APS/pyrrole molar ratio of 1:1 and NA/pyrrole molar ration constant at 0.6:1.

Naguib et al. in 2016 investigated another candidate polyacrymide (PAM) for fabricating MXene-base polymer composite [63] (Naguib et al., 2016). $Ti_3C_2T_x$ was first synthesized and intercalated with DMSO (dimethylsulfoxide) and then mixed with PAM in aqueous solutions. PAM is soluble in water and can form hydrogen bonding and DMSO-treated MXenes also aids in spontaneous co-intercalation of water between MXene layers. PAM/water

mixture was mixed with MXene/water mixture and then the mixture was vortexed and sonicated for 10–15 min until visual homogeneity was attained by the solution. In 2020, Xu et al. demonstrated fabrication of $Ti_3C_2T_x$/Polyaniline (PANI) by an important and effective technique, which improves the electrochemical performance of MXene-based electrodes. Fabrication was done under acidic conditions by chemical oxidative polymerization of aniline monomer on $Ti_3C_2T_x$ MXene layers. A total of 20 mL of 1M HCl solution containing 5 mmol aniline monomers was taken and 30 mg of delaminated $Ti_3C_2T_x$ powder was dispersed under ultrasonication for 30 min. The resulting mixture was then stirred in an ice bath and equal volume of 50 mg of APS in HCl solution was gradually added. The solution fully dispersed was left for polymerization via continuous magnetic agitation for 8 h at 0 °C. The final black-green product was washed, centrifuged, and vacuum dried for 24 h at 80 °C. The abundance of functional groups (such as –O and –OH) on MXene sheets, acts as nucleation centers for the deposition of aniline monomers. Polymerization process initiates at the functional groups after gradually adding ammonium persulfate (APS). Finally, the PANI will form a porous structure on MXene sheets, providing numerous channels for electrolyte ions to access the MXene sheets [64] (H. Xu, Zheng, Liu, Li, & Lin, 2020).

MXenes are impressive as additive materials for nanocomposite formation. There is a strong affinity for oppositely charged species aside from the facts that they also have high electrical conductivity. These factors help in active binding. The infusion of MXenes especially in polymer composites can reinforce the nanomaterial mechanically and make it tailorable for several applications.

*2.2. Exfoliation*

The synthesis process is not complete by etching and is therefore followed by major step known as the exfoliation and delamination (which is important for investigating and using the MXenes produced in the best possible way. The multilayered end product are stacked together by weaker hydrogen bond or VanderWaals bond compared to the strong M-A bonds which allows intercalation and delamination of MXene sheets into aqueous colloidal suspension.

Two primary ways of delamination have been explored, intercalation with large organic polar molecules like hydrazine, urea, isopropyl amine, tetrabutylammonium hydroxide (TBAOH), dimethyl sulfoxide (DMSO), and intercalation by cations like $NH_4^+$ and $Li^+$ along with sonication. The etching route and conditions mostly dictate the exfoliation method. Example, for HF-treated powders, intercalation of organic molecules or ions between the layers is preferred and necessary, which increases the space between the sheets and weakens the interlayer bonds making delamination easy. A schematic diagram of exfoliation process is provided below as Figure 5. The addition of bigger organic ions followed by vigorous shaking or sonication can provide independent 2D flakes.

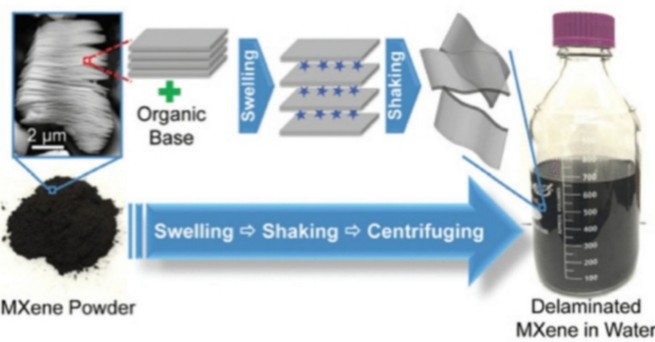

**Figure 5.** Schematic representation of delamination process of MXene. Reprinted with permission from ref. [65] (J. Zhu et al., 2017); Copyright 2017 Elsevier B.V.

Whereas, in some methods like LiF/HCl method, Li$^+$ ions are themselvesintercalated in the layers simultaneously during etching, which propels exfoliation process by vigorous shaking or direct sonication without additional intercalation [66] (L. Verger, V. Natu, M. Carey, & M. Barsoum, 2019). The typical requirements and steps of intercalation and delamination process are: a fitting solvent for the intercalation and material, the next step is mixing where the intercalant is incorporated between 2D sheets, occasional sonication depending on desired flake size and concentration, and a centrifugation step for isolating the delaminated material from the material, which is non-delaminated. After the entire process dispersed, stable 2D sheets will be present in the final colloidal solution. The first successful delamination was reported in 2013. We know that MXenes are –OH, –O, and –F terminated and so the exfoliated MXene produced is equivalent to $Ti_3C_2(OH)_xO_yF_x$ than the idealized $Ti_3C_2$ in pure form. So for simplicity, Mashtalir and colleagues used f-$Ti_3C_2$ [67] (Mashtalir et al., 2013) and it was intercalated with urea, HM (Hydrazine Monohydrate), HM dissolved in DMF (Dimethylformamide), and in DMSO (dimethylsulfoxide). DMSO successfully delaminates stacked MXene layers into single 2D sheets. It is to be noted that sonication is a required step after DMSO treatment. In a report presented in 2015 by Naguib and co-workers included a general approach to have large scale delamination of vanadium carbide and titanium carbonitrides and then generalizing the idea to other MXenes as well. They used large molecules organic base, namely tetrabutylammonium hydroxide (TBAOH), choline hydroxide, or n-butylamine (50–54% aq. And 10 mL solution) with 1 g of MXene followed by magnetic stirring for 2 h, 4 h, or 21 h [68] (Naguib, Unocic, Armstrong, & Nanda, 2015). The conclusion drawn was that the delaminated MXenes had a significantly low F-content. In another report published the same year, the failure of DMSO delamination of other MXenes led to a second route of amine assisted delamination employed on $Nb_2CT_x$ of MXene family. Freshly synthesized 1 g of $Nb_2CT_x$ was intercalated with 10 mL of i-PrA (isopropylamine) aqueous solution diluted with deionized water in a ratio of i-PrA:$H_2O$ as 1:4 and stirred with a magnetic stirrer for 18 h at RT. The suspension was then centrifuged for 10 min at 3600 rpm and the supernatant was decanted. This is a universal approach compared to DMSO-infused delamination and thus can be effectively applied to other MXene families as well [69] (Mashtalir, Lukatskaya, Zhao, Barsoum, & Gogotsi, 2015).

## 3. Results

Before mentioning the results of the aforementioned synthesis methods, a brief knowledge about the ways of obtaining the results is necessary.

### 3.1. Characterization

The most prominent characterization techniques used to analyze MXenes are X-ray diffraction (XRD) and scanning electron microscopy (SEM). For years, studies have been reported forother methods of characterization like FTIR, Raman spectroscopy, atomic force microscopy and plenty more according to the property of the synthesized MXene to be explored or based on MXene applications. Accurate characterization of any sample is important to validate the success of synthesizing the material. In case of MXene analysis, the purity of its precursor material MAX phase must be ensured as in most of the commercial or research samples available, multiple MAX phases can coexist that makes interpretation of data impossible. Taking example of a Ti-Al-C system, there is a possibility of the presence of both $Ti_3AlC_2$ and $Ti_2AlC$ phases together in the powder mixture, which is practically impossible to separate and study. So, first characterization of parent material is necessary to verify the changes obtained in derived material. According to a journal recently in 2020 [70] (Shekhirev, Shuck, Sarycheva, & Gogotsi, 2020), the best way to precisely confirm the purity of MAX phases is to employ XRD after applying significant texture to the powder. When a MAX phase fully transforms into MXene, except the (0001) peaks in XRD patterns all other peaks diminishes or completely vanishes. Also there is not only broadening of the (0001) peaks but also downward shift to lower angles and increase in the d-spacing of

layers, which indicates larger c lattice parameter (Figure 6). The first MXene synthesized by Naguib and colleagues (Ti$_3$C$_2$T$_x$), the XRD pattern showed the position of main peak shifted from 40° to10°. However, first step of MXene characterization is visual assessment as there is an obvious visual color change when MAX phases turns into MXenes, which is often overlooked. Depending on the composition and structure all MXenes have unique colors related to their optical properties while MAX phases in general are all grey colored (Figure 7).

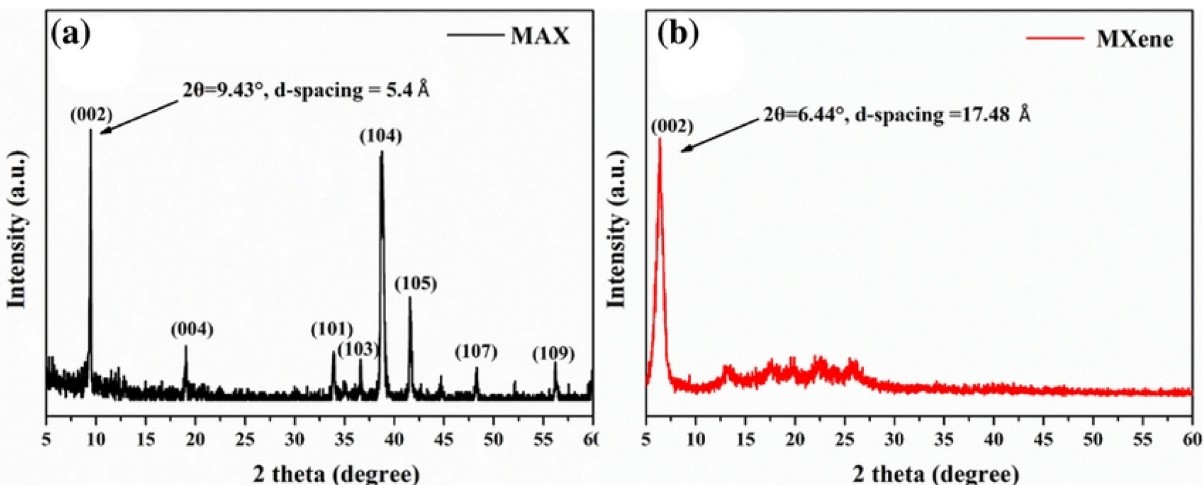

**Figure 6.** XRD data of (**a**) Ti$_3$AlC$_2$ MAX phase and (**b**) Ti$_3$C$_2$T$_x$MXene. It is clear that, except for the (0002) peaks, all other peaks in the MXene data almost vanished. Also, there is a downward shift of angle from 9.43° to 6.44° and a d-spacing increase from 5.4 Armstrong to 17.8 Armstrong. Reprinted with permission from ref [71] (Sheng et al., 2021). Copyright Springer US.

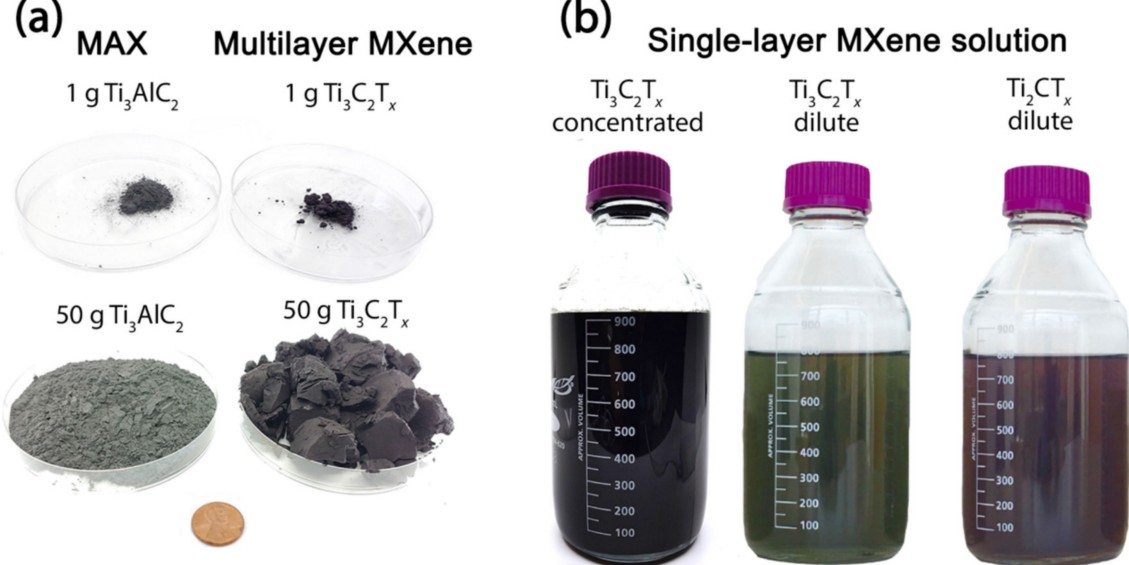

**Figure 7.** Color change shown by MAX phase and derived MXene. (**a**) Images of the parent Ti$_3$AlC$_2$ MAX phase used in synthesis of the 1 g and 50 g Ti$_3$C$_2$T$_x$samples. (**b**) Delaminated MXene solutions in 1 L bottles depicting their characteristic colors.Reprinted with permission from refs. [72] (Shuck et al., 2020) and [42] (Alhabeb et al., 2017). Copyright 2020 Elsevier Ltd.

XRD is very popular in MXene literature, with typically every article presenting both the MAX and MXene patterns, showing that the MAX phase was chemically converted to MXene or not. XRD can provide more information, leading to insight into MXenes

that cannot be attained in other ways especially, for instances where intercalation and delamination mechanism for a variety of organic and inorganic intercalants needs to be studied. However, XRD cannot probe the internal bonding structure and can only provide information about the presence of intercalants.

While assessing large scale delamination of MXenes using TBAOH [68] (Naguib et al., 2015), the XRD patterns for before and after treatment of TBAOH on $Ti_3CNT_x$ at different mixing times was observed. After 2 h treatment, there was an impressive shift in the 0002 peak of $Ti_3CNT_x$ from of 2θ 8.26° to 4.57° giving the increase in c lattice parameter from 21.4 Å to 38.6 Å. For 4h mixing the 2θ shift of 4.5° that corresponds to c-LP of 39.2 Å. Mixing for 21 h did not affect the c lattice parameter. This suggested that 4 h is the ideal and sufficient amount of time for treatment of the given MXene with TBAOH.

If we want a genuine indication that MXene is formed and also a visual depiction of the structure, then SEM is the choice. The accordion-like morphology that Naguib and colleagues talked about in their paper [23] (Naguib et al., 2014) can be visibly verified in SEM images which for a time being was taken as a sure indication that the MXene synthesis was successful. However further investigations has led to the fact that all multilayered MXenes does not share the accordion structure. This morphological transformation depends on the concentration of the etchant used. In a paper describing the guidelines of synthesizing titanium carbide Mxene [42] (Alhabeb et al., 2017), various concentrations of different etchants were used like HF and LiF/HCl solution. The SEM images are illustrated in Figure 8, whichshows that as HF concentration decrease, the accordion structure becomes less obvious or less prominent and both MAX phase and MXene images resemble to some extent. Similar conclusions were derived from the in-situ HF methods of synthesis and the optimized MILD method.

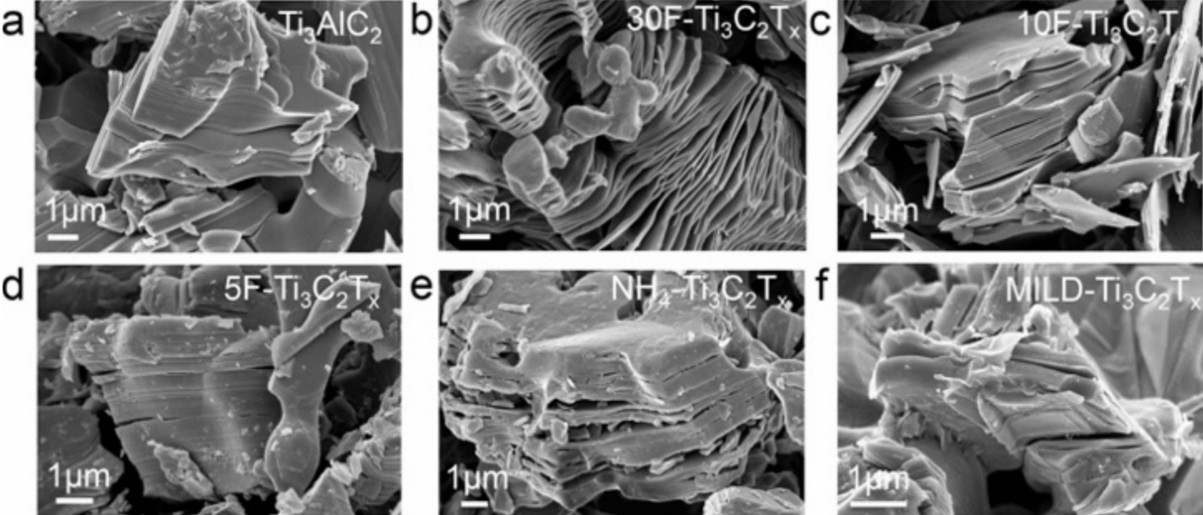

**Figure 8.** SEM images of MAX phases and MXenes synthesized with different etching routes, (**a**) $Ti_3AlC_2$ MAX phasewith compact structure. Multilayered $Ti_3C_2T_x$MXene obtained with HF concentration of (**b**) 30 wt%, (**c**) 10 wt%, and (**d**) 5 wt%. Only for 30 wt% the accordion-like morphology is observed for HF-etched powder. (**e**) Multilayered NH4-$Ti_3C_2T_x$.powder synthesized with ammonium hydrogen fluoride and (**f**) MILD method to synthesize $Ti_3C_2T_x$in 10M LiF/9M HCl, both showingnegligible opening of MXene lamellas, similarly observed in 5F$Ti_3C_2T_x$. Reprinted with permission from ref [42] (Alhabeb et al., 2017), Copyright American Chemical Society.

Both SEM and XRD are given importance on the basis of identifying the successful MXene production. However, both of them are powerful techniques and their data, both graphical and visual, can identify more than what is perceived recently. The contribution of both the methods in successful analysis of MXene by generating high resolution images

magnified up to five orders of magnitude and trustworthy data that is easily interpretable is outstanding and perhaps irreplaceable.

*3.2. Data of the Results of MXene Synthesis*

- *Etching using HF*
  - The first MXene synthesized Ti3C2Tx using the HF method followed Reactions (1)–(3) mentioned under the synthesis section. Reactions (2) and (3) shows the Mxene surfaces terminated by –OH and –F, respectively, in a simplified manner while in reality there may be a combination of both. The dominant reactions can be studied using DFT-based calculations. When the precursor was immersed in HF solution, bubbles (presumably H2) were observed, indicating the onset of chemical reaction.
    1. For the XRD pattern after ultrasonication, the reaction products in methanol for 300 s depicted significant weakening of the peaks and a broadened peak $2\theta = 24°$. FTIR (Fourier Transform Infrared Spectroscopy) confirmed the presence of –OH groups and the final result predicted was Ti3C2(OH)2.
    2. Along the basal plane, the elastic modulus was around 300 GPa of single exfoliated Ti3C2(OH)2layers.
    3. The presence of surface functional groups OH and F groups made the band structure semiconducting in nature.
  - This was obviously the first glimpse into MXenes and the impact that it can have in material science. As a new addition back then to the 2D material family, it must have been appealing to know more of the potential it possessed. The one thing that the onset of MXene experiments did is to shed some light and attention towards MAX phases. The top-down approach of synthesis seemed to be more convenient and important from a scientific point of view because it included their precursors' MAX phases. Now there are numerous parallel works going on about MAX phase to better understand MXenes. Also, synthesis of more MAX phases means the possibility of obtaining more MXenes with versatile properties.
- *Modified acid etching methods aimed at replacing HF as an etchant*
  1. NH4HF2 as an etchant—the use of this milder etchant was approved. From XRD data, the expansion of c lattice parameter from 18.6 Armstrong (Ti3AlC2) to 19.8 Armstrong (Ti3C2) was observed (the value matched previous recorded values). There was simultaneous intercalation of species like NH3 and NH4+, which further increased the c value to 24.7 Armstrong (25% larger than HF treated samples). The presence of F and O atoms between the layers was confirmed by XPS and EDX mapping in TEM. The intercalated Ti3C2-IC films attained 90% transmittance. Also, the absorbance of the films (independent of wavelength of light) was found to be linearly dependent on thickness. The Ti3C2-IC films showed greater resistivity than non-intercalated ones.
  2. LiF-HCl combo as an etchant—also successfully synthesized MXene. The result was clay-like paste rolled after hydrating to produce flexible free-standing films in a very short time compared to othersproduced by laborious techniques. Highly conductive when dry and could be given any shape when wet was the exciting feature. XRD and EDX data confirmed: the MXene is indeed formed, –O or –F containing surface functional groups, and the structure showing tight stacking of layers presumably due to cationic and/or water intercalation instead of having the claimed accordion-like morphology that HF-treated samples exhibit. Simple sonication yielded 45% by mass of dispersed flakes. Owing to the mild nature of the etchant, the flakes had larger lateral dimension and no nanometer-size defects frequent in HF-etched samples. The performance of the rolled up films as supercapacitor electrodes with H2SO4 as the electrolyte was excellent with volumetric capacitance up to 900 F/cm3 or 245 F/g.

- Although HF remains the choice as etchant, other etching agents can also effectively produce MXenes as well. However, which one of these will be suitable for industrial purposes is still a question. Also, we have to notice that different etching agents produce different results. Like LiF+HCl combo, it produced the 'clay-like texture', far from the accordion structure that was expected. These two etching chemical combinations are modified HF ones, which means they are not fluoride free, but are milder than HF. This makes handling them easier and also implies the nature of surface terminations the product may have. The in situ HF methods also causes an increased interlayer spacing by simultaneous intercalation of Li+ ions and waterthus weakening the interaction between bonds. This is an advantage especially in energy storage applications.

- *Fluoride-free synthesis methods*
  1. CVD process—fabricated ultrathin alpha-MO2C films of regular shapes like rectangles, triangles, hexagons, octagons, nonagons, and dodecagons. At 1090 °C for a growth time of 5 min crystal sizes of ≈10 μm and thickness of 3–20 nm was observed. By increasing the growth time to 50 min, very large thin crystals having lateral sizes of ≈100 μm was achieved at 1086 °C. The crystal structure was uniform and the surface was smooth like graphene. The films exhibited superconducting behavior, which was intrinsic thickness dependent ontemperature Tmin of 35K. Before the onset of superconducting transition, the films showed metallic character and very interestingly below Tmin the data indicated insulating behavior in the normal state. According to STEM observations, the ultrathin crystals were found to be defect-free.
  2. Urea glass route—XRD data with no other crystalline side phases and elemental analysis of the synthesized powders confirmed the fabrication of alpha-Mo2C and gamma-Mo2N of high purity. Theoretically predicted values of average sizes of Mo2C and Mo2N catalysts are ≈11 and 16 nm, respectively, which was in good agreement with the TEM analysis sizes 2–17 nm and 5–25 nm, respectively. Hydrogen evolution catalyzed by Mo2C in the KOH electrolyte had turnover frequencies (TOFs) of 0.5, 0.9, and 2.5 S−1 and onset potential of 176, 200, and 250 mV, respectively. For Mo2N, HER in KOH was catalyzed with TOFs of 0.07 and 0.7 S−1 at onset potential of 250 mV and 352 mV, respectively.
  3. Hydrothermal method—when Ti3AlC2 is hydrothermally treated with NaOH, Ti3C2Tx (Tx = –OH, –O) was obtained taking inspiration from Bayer process. XPS and XRD data gives a planar spacing of 24 Armstrong, which is larger than 20 Armstrong reported for HF fabricated Ti3C2Tx, suggesting Na+ interlamination. The SEM images show the compact layered structure. Ti3C2Tx film electrode (52 μm thick) without any—F terminations in 1 M H2SO4 possessed a gravimetric capacitance of 314 F/g and volumetric capacitance of 511 F/cm3 at 2 mV S−1. Compared to HF-Ti3C2Tx, capacitance is higher by ≈214% and for LiF + HCl—Ti3C2Tx clay the value surpasses by ≈28.2%. With another hydrothermal route using (NaBF4, HCl) as the etching agents, two kinds of MXenes Ti3C2 and Nb2C were prepared. The resulting Ti3C2 show adsorption to cationic dye (MB), the adsorption performance being better than the traditional HF etched samples.
  4. Electrochemical method—Ti3C2 produced by electrochemical corrosion of Ti3AlC2 in binary aqueous electrolyte was studied for capacitive behavior and the observations stated the areal and volumetric capacitances to be 220 mF cm−2and 439 F/cm3, respectively at a scan rate of 10 mV S−1.

     - HF hasa corrosive nature, which often hampers the yield of the product. Out of the four methods listed above, the hydrothermal method is the successful one in terms of producing better yield and enhanced properties compared to the HF etched products. Although one thing should be kept in mind that if we talk about synthesizing only, then the fluoride component has to

play an important part. Recently, a study suggested that mechanical stress can also break the A layer bond in MAX phases, which if proven can be a huge leap towards industrial purposes. Mechanically breaking the bond will be so cost effective and harmless that it can change the entire picture. However, it is yet to be determined whether this concept will be true to every MAX phase or not because of the varying bond strength [73] (Guo, Zhu, Zhou, & Sun, 2015). Also, Mo2C fabricated by CVD process showed thermal and chemical stability in solvents like ethanol, isopropanol, and HCl. The strength of Mo2C is 20.8 GPa, which is almost comparable to a monolayer of MoS2. This makes the Mo2C MXene a choice for mechanical applications.

- *Molten fluoride salt etching method*
  1. The mixture of molten fluoride salts produced the nitride base MXene Ti4N3Tx. Spin polarized density functional theory (DFT) calculations were performed and according to density of states at the Fermi level are metallic for both bare and functionalized Ti4N3. Bare Ti4N3 is magnetic, while introducing surface termination groups lowers the total magnetic moment in unit cells. The OH introduction dramatically lowered the magnetic moment from 7.0 μB to zero.

     - The most studied MXene is titanium carbide. This was a novel material discovered using the molten salt method, which had not been reported earlier. The study was made between terminated and non-terminated Ti4N3 monolayers by using spin-polarized DFT calculations. The electronic structure is explained via the partial density of states (PDOS) of bare Ti4N3 and Ti4N3Tx with all the possible terminations. The bare MXene shows the same metallic character as the carbide MXenes, while introducing surface terminations tends to lower the DOS at the Fermi level. However, among the surface terminations, only –F terminated Nitride MXenes were theoretically calculated for DOS in all previous research works. Considering –OH and –O terminated ones in this paper, it was found out that comparatively –OH terminated ones are the least energetically favorable and –O terminated MXenes are most favorable. The fact that the magnetic moment can be tuned by tuning F/OH and F/O ratio opens up a completely new field of research of MXene applications. Another nitride-based MXene was synthesized (Ti2NTx) and was tested for SERS activity [30] (B. Soundiraraju & B. K. George, 2017). SERS sensitivity was confirmed in Ti2N which makes it the candidate for developing self-standing flexible SERS substrates. This is a completely different direction of application in detecting traces of explosives and later can be expanded to capture other poisonous contaminants in the environment.

- *MXenes prepared from other precursors than Al-based MAX phases*
  1. 2D Zr3C2TxMXene was prepared for the first time from parent ternary-layered Zr3Al3C5, which was a fresh approach to obtain MXenes outside of Ti-C systems. It was found out that all of the three functionalized Zr3C2Tx MXenes (T = OH, F, O) exhibit metallic behavior with Zr3C2T2 having the strongest mechanical strength of calculated elastic constants as high as 392.9 GPa. It showed greater stability in argon atmosphere or complete vacuum than the titanium carbide.
  2. Synthesized Mo2C from gallium-based laminate Mo2Ga2C is the first instance of non-Al containing MAX phases used for fabrication.

     - This is a no brainer that Ti3C2Tx is the most studied MXene, and also the Al-based MAX phases are the most wanted precursors. However, shifting from the conventional ones brings unknown properties forward, which may find a unique kind of application. Not much experimental data is present regarding this idea as synthesis of non-Al MXene itself is in progress. The

main thing about all these attempts is to prove that we can customize our MXenes as per our requirements.

- *MXene-Polymer composites*

  1. For Ti3C2Tx/PVA composite, the tensile strength comparatively improved by 34%. While the mechanical strength of Ti3C2Tx (3.3 μm thick) was found to be 22 + 2 MPa or 22 − 2 MPa with a Young's Modulus of 3.5 + 0.01 GPa or 3.5 − 0.01 GPa, the strength of Ti3C2Tx/PVA (PVA loading of 60 wt%) film enhanced four times to 91 + 10 MPa or 91 − 10 MPa. Ti3C2Tx/PDDA composites did not have any impressive or enhanced capacitate effect, but combination of PVA and KOH gel electrolyte showed significant increase in volumetric capacitance: 528 F/cm3 at 2 mV/S and 306 F/cm3 at 100 mV/S.

  2. For PPy/Ti3C2Tx composite, XRD and TEM data confirmed pyrrole intercalation in between layers. Composite exhibited volumetric capacitance of ≈1000 F/cm3 and capacitance retention of 92% after 25,000 cycles.

  3. The pyrrole composite was further improved using 1,5 naphthalene disulfonic acid (NA) and cetyltrimethylammonium bromide (CTAB). Specific capacitance calculated for pure PPy, Ti3C2Tx/PPy, NA-Ti3C2Tx/PPy, and NA-CTAB-Ti3C2Tx /PPy composite was 128, 266, 318, and 437 F g−1, respectively. This is clear that the composite yielded a higher specific capacitance. NA-CTAB-Ti3C2Tx/PPy composite showed good electrochemical properties with capacitance retention ratio of 76% after 1000 cycles compared to only 50% for the pure PPy electrodes.

  4. In 6 wt%MXene/PAM nanocomposite films, SEM, TEM, and XRD confirm the presence of well dispersed nanoflakes of MXenes. The conductivity increased significantly to $3.3 \times 10-2$ S m−1 for 1.7 vol% MXene loaded films by increasing the MXene loading.

  5. MXene/PANI composite showed excellent electrochemical performances including high specific capacitance of 556.2 F g−1 at 0.5 A g−1, outstanding cycling stability of almost 91.6% retention after 5000 cycles and good rate capability with 78.7% capacitance retention at 5 A g−1.

     - MXene-based composites are the next step evolution in material science. They are leaving their impact on energy storage applications as well as in environmental applications, for example, water purification and sensors. Sure enough MXenes are considered as good electrode materials, but as mentioned above, the capacitate retention is more in case of MXene/PANI substrates than both pure MXene and PANI. So with the high electrical conductivity of MXene, the pseudocapacitive nature of PANI is added, which explains the good rate performance. MXene-PVA composite shows four times more mechanical strength than conventional titanium-based MXene. On the flip side, it can be said that MXenes are suitable nanofillers for obtaining high performance polymer composites. In a study, it has been reported that MXene-based composites are superior as electrodes in supercapacitors, Li-ion, Li-S, and Na-ion batteries. The large surface area of MXenes does encourage deposition of charge during each cycle of charging and discharging in supercapacitors. However, their energy density is limited and also the longevity is questionable. So, reinforcing them with polymers constitutes double layers for pseudocapacitive effect that improves the performance drastically. MXenes degradation is a big problem and can be rectified by forming composites that act as stable matrices for holding the structures and giving a good cycle stability [74] (J. Yang et al., 2019).

## 4. Discussion

The beginning of this extraordinary discovery gave the research area some intriguing results. Since graphene was hyped for all the right reasons, similarities were drawn between

MXene and graphene (even the nomenclature indicates the inspiration: 'Mx' from MAX phases and 'ene' from graphene):

1. Resemblance of the multilayer sheets to exfoliated graphite.
2. The ability to form rolled scrolls similar to sonicated graphene scrolls.

Although having structural similarities, there were glaring differences also specially in the properties experimentally obtained. For example, while the elastic modulus value of 300 GPa of the MXene layers was lower than that of graphene [The surface terminations indicate better bonding polymer matrices and can be used as fillers in polymer composites. Also there is a probability of bonding rigidity of $Ti_3C_2$ layers greater than graphene and the MXene sheets shows greater stability than graphene sheets. It is also worth discussing that the synthesis routes morphed over the years illuminated a great deal about the properties of MXenes. With each experimental endeavor, reaction conditions were tuned and that resulted in enhanced features. For instance the MXenes synthesized by HF etching method had the iconic accordion like structure. But the methods in which there was simultaneous intercalation and etching, the resulting MXenes had stacked uniform structure rather than the expected accordion morphology. From mechanical features like flexibility, bending rigidity to electronic characteristics like increased volumetric capacitance, smooth charge transport has been found to be varying constantly according to demand and still has a lot of room for improvement. From the results of various research works, it is clear that the role of surface terminations is vital for determining the applicability of MXenes in the best possible way; like the lowering of the magnetic moment when surface groups were introduced compared to bare MXene.

*4.1. Properties*

4.1.1. Mechanical Properties

Strength and elastic properties of self-standing MXenes are mainly dependent on factors like (i) presence of surface functional groups; (ii) number and thickness of layers.

1. Effect of surface terminations

It was found that MXenes with –O terminations were highly stiff, but for –OH and –F terminated flakes, the elastic stiffness was very low comparatively [75] (Caffrey, 2018). However, comparing with no surface terminated MXenes; the ones having the terminations were largely flexible. According to Guo et al. mechanical response of 2D $Ti_2C$ was recorded along x and y direction [76] (Guo, Zhou, Si, & Sun, 2015). Conclusions were drawn that 2D $Ti_2C$ is elastically isotropic, having Young's modulus values of 620 GPa and 600 GPa in both directions. However, upon functionalization, although there was a reduction in Young's modulus, it could still withstand larger strain than bare $Ti_2C$ and even graphene. Presence of –O functional group creates a buffer layer that slows the tearing of Ti layers by increasing the value of critical strain where $Ti_2C$ collapses. First principles DFT calculations of 2D $Ti_2CT_2$ and $Ti_3C_2T_2$ also shows that high dependence of strength on surface termination groups. Comparing the stiffness, it was found out that the –O terminated MXenes shows more elastic stiffness than –F and –OH terminated ones [30] (B. Soundiraraju & B. K. George, 2017). A different literature also points out that $Ti_2CO_2$ compared to bare $TiC_2$ and graphene has higher percentage of critical strains [77] (H. Zhang et al., 2017).

2. Effect of number of layers and thickness of layers

The second factor worth discussing is the number of layers 'n' in $M_{n+1}X_n$ having a great influence on mechanical properties of MXene. A realization of hydrogen bonding between the layers of MXene manifested experimentally when elastic modulus values of bilayer $Ti_3C_2T_x$ flakes were found to be twice the values obtained for single layer MXenes [78] (Lipatov et al., 2018). The study of elastic constant and young's modulus by Yorulmaz et al. showed that among Ti-C system, the thinnest $Ti_2C$ has the highest Young's modulus [79] (Yorulmaz, Özden, Perkgöz, Ay, & Sevik, 2016). It was also shown through MD calculation on $Ti_{n+1}C_n$ that Young's modulus significantly increases with decreasing

layer thickness [80] (Borysiuk, Mochalin, & Gogotsi, 2015). Recent experimental endeavors have shown that there is enhancement of mechanical strength of MXenes infused with polymer matrix as composites or carbon nanotubes.

Other factors that affect the mechanical strength are mass of transition metal, doping, varying F/O ratio, intercalation of ions and electrolytes mostly theoretically studied using DFT calculations [81] (Ibrahim & Mohamed, 2020). The doping affects the elastic properties of MXenes according to theoretical calculations. In a study, B andV atoms were doped in Ti and C lattice points. The doping of V atoms did not make much difference, but there was noticeable decrease of planar Young's Modulus value, thus improving the elastic properties. The weak bonding in Ti-B MXenes compared to conventional Ti-C MXenes is the cause of reduction of stiffness [82] (Chakraborty, Das, Nafday, Boeri, & Saha-Dasgupta, 2017). Comparisons are also being made to graphene, h-BN, 2D $MoS_2$, and SiC with the B atom-doped MXene $Ti_2(C_{0.5}B_{0.5})$, the later showing a stiffness value 3.1 times higher than the other 2D materials listed [83] (Andrew, Mapasha, Ukpong, & Chetty, 2019). For MXenes with mixed surface terminations especially –F and –O, the elastic properties can be tuned by manipulating the F/O ratios. In a study, the five most thermodynamically stable structures having F/O ratios of 1:2, 2:1, 5:4, 7:2, and 17:1 were experimented on. The conclusion was that by increasing the F/O ratio there was a considerable decrease in Young's modulus that resulted in better mechanical properties [84] (Almayyali, Kadhim, & Jappor, 2020). In addition to elastic properties, MXenes show higher bending rigidities than graphene monolayers [81] (Ibrahim & Mohamed, 2020).

### 4.1.2. Electronic Properties

Perhaps the most interesting property of MXenes is its electronic property. Extensive DFT calculations show that bare monolayer of MXene are predictably metallic in nature, having a high electron density near the Fermi level [85,86] (Naguib et al., 2011b; Shein & Ivanovskii, 2012). Although this prediction was for MXenes without any surface termina-tions and un-terminated MXenes have not been synthesized yet to confirm this prediction. However, in theory, the surface functional groups actually dictate the electronic properties of MXenes indicating sensitivity of their work functions towards the type of functional groups [87] (Khazaei et al., 2013b). The –OH terminated MXenes show extremely low value of work function equivalent to or less than 2.8 eV, which is lower than Sc known for having the lowest value among metal elements. On the flip side, –O terminated MXenes show work function values even larger than Pt, which has the highest work function among elemental metals. The work function values of the MXenes with –F terminations lie between the –OH and –O terminated equivalents. After surface functionalization, the drastic change in work function values occurs due to change of surface dipole moment induced by the functionalization itself [88] (Liu, Xiao, & Goddard III, 2016). So, apart from metallicity, MXenes also exhibits semiconductivity and can be used in topological insulators [89,90] (Khazaei, Ranjbar, Arai, Sasaki, & Yunoki, 2017; Khazaei, Ranjbar, Masao, & Yunoki, 2016). The metallic nature is manifested in MXene as high electrical conductivity, which makes them perfect for many applications, especially as electrode materials in batteries and su-percapacitors. The conductivity ranges from less than 1 S $cm^{-1}$ to thousands of S $cm^{-1}$ depending on their synthesis routes and delamination technique. HF-etched $Ti_3C_2T_x$ had low conductivity of 2 S $cm^{-1}$ due to more defects and higher F terminations, whereas a conductivity of almost 1500 S $cm^{-1}$ was obtained in LiF+HCl-etched $Ti_3C_2T_x$ sample [91] (Hu et al., 2015). Most reviews support the metallic or semi metallic nature of MXenes and there is less experimental evidence on MXenes potentially being semiconductors. Some early investigations computed the band gaps in the range of 0.24 eV to 1.8 eV of MXenes $Ti_2CO_2$, $HfCO_2$, $ZrCO_2$, and $Sc_2CT_2$; T=OH,O,F. It was obtained as a result that especially in the case of $Sc_2CT_2$ that for –F and –OH terminations, the band structures were comparable and similar to the –O terminated ones. This proves that surface functionalization has a role in categorizing MXene as a semiconductor [87] (Khazaei et al., 2013a). Wang et al. shed some light on the band gaps of Ti, Zr, and Hf-based MXenes in accordance with their

similar electronic configurations [92] (Y. Wang, Xu, Hu, Ling, & Zhu, 2020). Owing to the semiconductive nature of MXenes, they can be used for photocatalytic activities. The blend of electrical and optical properties may give the most useful application of MXene in an environmental field till now. For catalytic applications, there is a clear correlation between the optical and electronic properties. MXenes absorb light in the visible region, which explains the hydrogen production by water dissociation using only sunlight. Sinopoli and group documented the electro/photocatalytic properties of MXenes. Theoretically, $Ti_2CO_2$, $Hf_2CO_2$, and $Zr_2CO_2$ have band edge positions of 0.24 eV, 1.0 eV, and 0.88 eV, which matches with the redox potentials of general water splitting. So, these MXenes are capable of producing visible light-driven photocatalysis [93] (Sinopoli, Othman, Rasool, & Mahmoud, 2019). Aiming towards degrading or adsorbing organic pollutants like dye in water. Normally used $TiO_2$ nanoparticles show good photocatalytic activity and degrade organic molecules effectively. However there was a lack of redox potentials on surface when exposed to light, so an alternative or reinforced material was sought out. –F and –OH terminated MXenes behave as negative particles in water, which adsorbs the cations far more effectively than the anions. In a study, it has been shown thatbinary-MXene alloy systems like $Ti_{2(1-x)}Zr_{2x}CO_2$, $Ti_{2(1-x)}Hf_{2x}CO_2$, and $Zr_{2(1-x)}Hf_{2x}CO_2$ display good photo-catalyticbehavior. $Ti_{2(1-x)}Zr_{2x}CO_2$ is predicted to be the most effective photocatalyst in water splitting with the x value 0.2778. So, stoichiometric changes in MXenes to form alloys can tune photocatalysis performance [94] (Wong & Tan, 2018). MXenes also contribute in hydrogen evolution reaction (HER) and oxygen evolution reaction (OER) at cathode and anode, respectively, producing $H_2$ and $O_2$ molecules. This is why MXenes are considered for water electrolysis. Higher surface areas due to 2D structure, hydrophilicity, and electrical conductivity are the main factors for which MXenes are good catalysts or serves as catalyst supports. Comparing with other 2D materials like $MoS_2$ and carbon nanotubes, MXenes possess higher charge transfer abilities. In a recent study exploring the new horizon in MXene catalysis, besides HER and OER, $CO_2$ reduction, $N_2$ reduction reaction are also discussed in detail. Obviously, all the works till now have mostly been theoretical calculations and need more experimental backup for validation [95] (Morales-García, Calle-Vallejo, & Illas, 2020).

### 4.1.3. Optical Properties

Transparency, absorbance, and behavior when exposed to a range of wavelengths can be summarized under the optical properties of MXenes. Bare $Ti_2C$, $Ti_3C_2$, $Ti_2N$, and $Ti_3N_2$ MXenes were tested for reflectivity, absorption spectrum, and energy loss function [96–98] (Bai et al., 2016; Lashgari, Abolhassani, Boochani, Elahi, & Khodadadi, 2014; Ying, Dillon, Fafarman, & Barsoum, 2017). As a result, the Plasmon energies of the above-mentioned structures are 10.00 eV, 10.81 eV, 11.62 eV, and 11.38 eV, respectively. It was also shown that if the energies went down below 1 eV, there could be 100% reflectivity inferring the capability to transmit electromagnetic waves. When spin cast films produced from MXenes are thin enough they can be transparent to the visible region of spectra. Spin cast thin films of $Ti_3C_2T_x$ for example have competitive transmission of 97% per nanometer thickness with transmission in a single thin graphene layer, which is equal to 97.7% [99,100] (Dillon et al., 2016; Raveendran-Nair et al., 2008). The absorbance of $Ti_3C_2T_x$ and $Ti_2CT_x$ thin films is comparable at a wavelength of 550 nm [98] (Ying et al., 2017), while $V_2CT_x$ are found to be doubly transparent [101] (Ying, Kota, Dillon, Fafarman, & Barsoum, 2018). Also, intercalation of organic molecules or cations can hugely alter the absorbance capacity of the films. The high transmittance and electrical conductivity combo of $Ti_3C_2T_x$ thin films propels them to be used in optoelectronics. Intercalating with TMAOH has shown to increase their transmittance by 20%. Again, it is worth noting that etching of films by $NH_4HF_2$ bestows higher transparency in films than that fabricated with traditional HF etching.

## 5. Applications and Recent Works

The exclusive properties of MXenes are indication enough that they can have their own spot in manufacturing industries and device applications. The electronic, optical and mechanical properties discussed above have their individual areas of applications for MXenes and numerous works are going on to overcome the challenges to produce optimum results. The passing of the experimental stage is crucial though to make them commercially successful and so we need to find out the sectors where they can have the most use. Through the years there are several ways MXene is being researched on in order to get a picture of their usability. Having the unique combination of properties like mechanical properties contributed by the transition metal carbides/nitrides, high electrical conductivity; hydrophilicity due to functionalization of surfaces; ability to absorb electromagnetic waves efficiently and high negative zeta potential enabling formation of stable colloidal solutions are the explored genres till date and can give way to various applications. The pie chart depicts the sectors of applications (Centre pie chart) Figure 9. On the second ring the years when they were first explored is shown. It appears that MXenes have found their niche in energy storage applications and recently in environmental sectors as well. As per the data collected, the initial stages of experiments were focused on the structural mapping of the MXenes. The energy storage field was the first thing that was sought for regarding MXenes and remains the large portion of MXene activities. Recently other fields are also considered like the Biomedical and Photo catalysis applications in the last couple of years are gaining importance rapidly. The energy sector includes use of MXenes in electrochemical capacitors, micro-super-capacitors, Li-ion batteries and others. Back in 2012, Naguib and team first experimented on MXene based Li-ion battery. It was established that due to a higher exposed surface area and easy intercalation between weakly bound MX layers, the capacity was 5 times higher than that of $Ti_2AlC$. Exfoliated $Ti_2C$ demonstrated a stable capacity of 225 $mAhg^{-1}$ at C/25 rate. This showed that MXenes can be used for energy storage other than the bulky counterparts [25].

After that work, other research teams also lead review studies aiming at establishing MXene as a superior electrode material. Aierken and his fellow researchers produced a research paper in 2018 about MXene/graphene heterostructures in Li batteries. The physical properties of the system in consideration were derived from DFT calculations. The MXene/graphene bilayer vertical heterostructures ($M_2CX_2$ + Gr, where M = Ti, Sc, V and X = OH, O) were investigated with respect to the strongest Li binding sites. $Ti_2CO_2$ + Gr and $V_2CO_2$ + Gr emerged as promising systems for Li intercalation. There was 100% Li intercalation in these two systems, which means all the sites were occupied resulting in an open circuit voltage of 1.49 V for $Ti_2CO_2$ + Gr and 1.93 V for $V_2CO_2$ + Gr bilayer. The molecular weight of the heterostructure bilayer was lower than exposed MXene bilayer and so offered higher storage density [102] (Aierken, Sevik, Gülseren, Peeters, & Çakır, 2018). Tang et al. also gave a review regarding MXene's suitability as an electrode material for storing energy. The vast review covered the use of simple MXene as well as MXene-based composites in Li-ion batteries, Na-ion batteries, Li-S batteries, supercapacitors, hybrid-capacitors, and sodium-ion hybrid capacitors in a systematic way [103] (Tang et al., 2018). The results pointed out the pros and cons of MXene use in batteries and also the amount of experimental work needed to further increase the scope. The paper also pointed out the fact that the stacked nature of MXenes makes it capable of energy storing via intercalating charges in between. So, if the interlayer distance can somehow be increased without hampering the stability of the structure, then the capacity to store more charge will increase. A recent review in 2020 also discussed the same topic, solely focused on the effect of the interlayer distance in MXenes [104] (Garg, Agarwal, & Agarwal, 2020).

The interlayer distance is very crucial and so is the choice of cations to be intercalated. Larger cations can lower conductivity thus affecting the transfer of charges and ions. By intercalating potassium ion there was a small increase in interlayer distance contrary to intercalating lithium, sodium, magnesium which led to electrode contraction. The lack of experimental evidence regarding MXene-polymer composites for energy storage purpose

makes it difficult to compare and determine which one is better. Lin et al. also published a paper regarding MXenes as high rate electrodes in both aqueous and non-aqueous electrolytes as well as in organic electrolytes containing metal ions [105] (Z. Lin, Shao, Xu, Taberna, & Simon, 2020). The choice of solvent is equally important as the surface terminations of MXenes. The electrochemical performance in acidic aqueous electrolytes shows an ultra-high volumetric capacitance of 1500 Fcm$^{-3}$ titanium carbide serving as electrodes [13] (Lukatskaya et al., 2013). This value is definitely greater than what offered by conventionally used porous carbon and its derivatives. The only limitation was the narrow potential window resulting from the rapid oxidation of MXene at higher negative potential significantly lowering the energy density. That is why the focus shifted to non-aqueous electrolytes (organic or ionic) having a stable potential window ranging from 3V to higher values. This can change the entire picture but the specific capacitance remains limited in this case compared to the aqueous electrolyte which was exceptionally higher. This is due to the shrinkage of MXenes reducing the interlayer distance and thus cutting off the access to cations or anions. Thus the possibility remains of finding a point of stability for MXenes to avoid restacking and dictating a specific design with tuned interlayer spacing. In 2021, a perspective was laid forward by Balach and Giebeler about the progress and use of MXenes in Li-S batteries [106] (Balach & Giebeler, 2021). The paper theoretically develops to extend applications in micro scale industry and generate autonomous high power reservoirs without compromising the size by significantly enhancing the Li-S battery features using MXene properties. The idea of miniature Li-S batteries is not yet entertained due to the shuttle effect (the formation of polysulfides in the electrolyte while reaction that can poison the anode or depletion of cathode via rigorous redox reactions). Also it is important to establish the superiority of Li-S batteries experimentally over the already existing Li-ion batteries. The idea of MXenes as electrodes curbing the shuttle effect while meeting the criteria of reliability fits perfectly. To make the electrodes last longer in a manner of self-healing via making the broken bonds during the ongoing reaction, the proposal to use MXene based polymer composites is also coined. The ambitious study poses solid questions and opens many areas of research of synthesis of MXenes not just in its simple form but with other binders serving as stable matrix.

Microsupercapacitors are now most probably the most exciting devices in energy storage applications. The era now and upcoming will always strive to miniaturize and elongate the lifetime of batteries without constantly having to replace or replenish them. Electrochemical capacitors or supercapacitors are emerging as alternatives for batteries as they are theoretically meant to have unlimited lifetime and provide high power density. The only issue is their size and bulky nature, which is not suitable for micro devices. Thus, the urge to shift towards microsupercapacitors (MSC) is rational and is being adopted to be integrated with electric circuits in a micro level. According to a review, recently the MSC available is of two types: microelectrodes arranged in an array in microscale sizes and thin film electrodes having sandwich-like structure and thickness less than 10 μm [108] (Beidaghi & Gogotsi, 2014), [109] (S. Zheng, Shi, Das, Wu, & Bao, 2019). Through the years, carbon and its derivatives are used for constructing microsupercapacitors like carbon-derived carbon [110] (Chmiola, Largeot, Taberna, Simon, & Gogotsi, 2010), carbon nanotubes [111] (J. Lin et al., 2012); graphene [112,113] (J. Liang, Mondal, Wang, & Iacopi, 2019), (G. Zhang et al., 2018) and others [114–116] (Kim, Hsia, Carraro, & Maboudian, 2014), (Kim et al., 2014), (Pech et al., 2010), (S. Wang, Hsia, Carraro, & Maboudian, 2014). However, they provided low-energy density that overshadowed their high conductivity. The alternate use of pseudocapacitive materials like transition metal oxides and conducting polymers showed poor conductivity [117–120] (Kurra, Hota, & Alshareef, 2015) (Meng, Maeng, John, & Irazoqui, 2014), (Augustyn et al., 2013), (Choi, Blomgren, & Kumta, 2006). This is why the MXenes with their 2D morphology and experimentally tunable surface terminations are claimed to be a better replacement. The surface terminations actually install the property of hydrophilicity in MXenes that in turn significantly influence the Fermi level densities. Peng et al. in 2016 fabricated Ti$_3$C$_2$T$_X$ MXene MSC by spray coating

technique and painting a PVA/H$_2$SO$_4$ gel electrolyte. The volumetric capacitance was found to be 356.8 Fcm$^{-3}$ at 0.2 mA cm$^{-2}$ along with a high areal capacitance of 27.3 mF cm$^{-2}$ at 20mVs$^{-1}$ [121,122] (Y.-Y. Peng et al., 2016). (Jiang et al., 2019) found out that Ti$_3$C$_2$T$_X$MXene exhibits superior capacitate nature comparable to commercially available 4 mF capacitor. This shows that the bulky electrolytic capacitors are replaceable with MXene based MSCs [123] (Jiang et al., 2019). Li et al. fabricated double-sided MSC's (DMSC) using MXene ink attached to the window of 7.2 V working potential in various series and parallel configurations. It was seen that by decreasing the electrode gap, the capacitance rose steeply with 10 μm gap showing the highest volumetric capacitance of 308 Fcm$^{-3}$ at 5 mVs$^{-1}$ [124] (Q. Li et al., 2020). Another study by Sharma and Rout illustrates the electrochemical properties and also the various fabrication techniques of microsupercapacitors pointing MXenes as a promising material. MXene-based hybrids and MXene/CNT-based MSCs are also discussed in this paper along with a detailed table reporting other hybrids too [125] (Sharma & Rout, 2021). There is a clear lack of experimentation to further enhance the use of MXene in MSCs as this needs careful regulation and optimization of working electrode materials. Also, the ambition to get higher areal capacitance and higher energy density should be the main point to keep in mind while developing the electrodes.

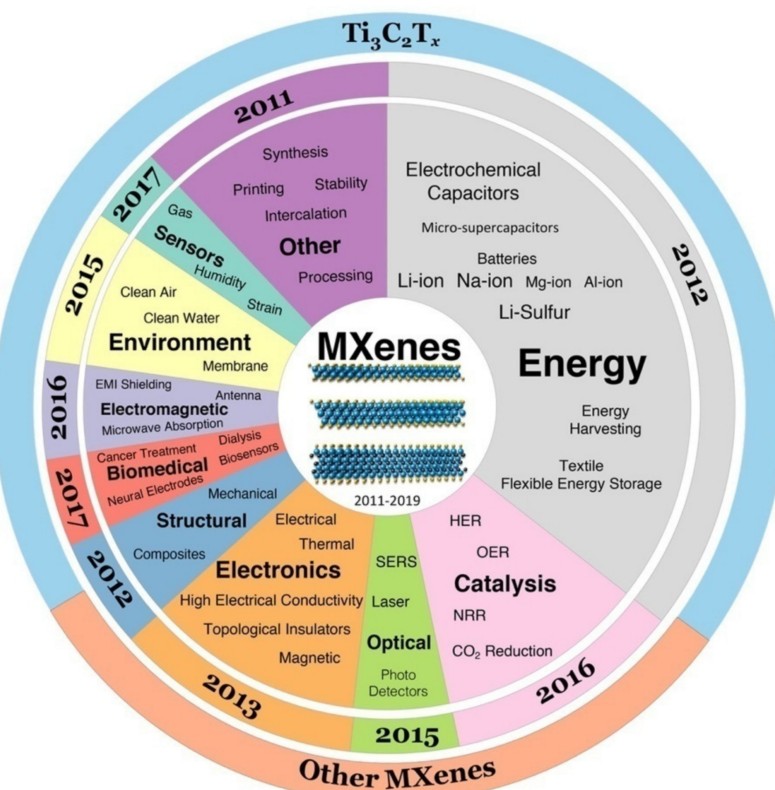

**Figure 9.** Mxene applications: The pie chart depicts the MXene applications in various sectors explored. Center pie chart depicts the no. of publications made for the applications mentioned. The middle ring shows the year of onset of those publications; i.e., the breakthrough works with most impact. The last ring shows the ratio of explored Ti$_3$C$_2$T$_x$ to other MXenes. Adapted with permission from [107] (Gogotsi, 2019). Copyright © 2019 American Chemical Society.

Photocatalysts originated in the 20th century only propelled by the need of developing environment-friendly technologies and devices. The prospects of clean energy, nontoxic wastes, and environmental wellbeing are the prime concern of any new advancement regarding technology and photocatalysts are able to fulfill all these criteria. Their nature is semiconducting and when light falls on them and the electrons from valence band leaps to the conduction band leaving behind a hole at initial position. The process is guided

by oxidation and redox reactions that need to be understood and explored more. This migration of the charge carriers' photogenerated by the incident light to the catalyst surface is the main step, also the rate determining step of the entire reaction. The inhibition of the recombination of the charge carriers aids in the photocatalysis process [126] (B. Wang, Zhang, & Huang, 2017), [127] (Y. Li, Jin, Zhang, & Fan, 2019). The reaction mechanisms typically involve photocatalytic hydrogen evolution reaction (HER), reduction reaction of $CO_2$ ($CO_2RR$), and degradation reaction of photocatalysts. Veering towards nanomaterial specially in two dimensions, the thickness goes down compared to the three-dimensional ones, which makes the Fermi level discrete and the energy gap wider, enabling more excitation of photons [128] (X. Li et al., 2021). A recent review by Kuang and team displayed the MXene-based photocatalysts showing light on the use of MXenes and their role in photocatalysis. Being a 2D material puts MXenes as the tool to be used in photocatalysts. The paper establishes other factors also in a very organized way that proves the usability of MXene in this type of emerging technology. The abundance of surface functional groups and large surface area enhances MXene's possibility and makes them infuse the photo activated separation of charge carriers at the same time, acting as stable support or surface for the reactions to occur. MXenes also limits the size of the photocatalysts and thus has been used in various photocatalyst-based applications like water splitting, nitrogen fixation, and carbon dioxide reduction [129] (Kuang, Low, Cheng, Yu, & Fan, 2020). In 2016, a study came out claiming MXene as a promising photocatalyst for water splitting [130] (Guo, Zhou, Zhu, & Sun, 2016). All three types of transition metal carbides ($M_2C$, $M_3C_2$, $M_4C_3$) and the analysis were done by XPS and EDS. A varied range of transition metal elements were experimented with, out of which 2D $Zr_2CO_2$ and $Hf_2CO_2$ was successful in water splitting. It has been already established that $Ti_3C_2T_X$ is the most used MXene as photocatalyst in HER [131] (Takanabe, 2017), [132] (Kudo & Miseki, 2009). The hydrophilicity promotes the reaction and the Gibbs free energy aids in the adsorption of hydrogen. $Ti_3C_2T_X$ acts as a base catalyst or a co-catalyst enhancing the performance of the base catalyst by doubling yield of $H_2$. A single 2D flake of $Ti_3C_2T_X$ is more suitable for this job; however, extracting a 2D flake is complex. One major setback is instability of MXenes for applications in aqueous media. MXenes are not highly resistant to oxidation despite ambient conditions and is considered to be thermodynamically metastable. $Ti_3C_2T_X$ can even oxidize in normal air under atmospheric pressure with Ti turning to $TiO_2$ leaving behind a sandwich-like structure of carbon layers and $TiO_2$ layers. Also, it has been determined that $Ti_3C_2T_X$ contains Ti defects intrinsically that aid in further instability. This instability contradicts the accepted behavior of long lifetime in photocatalysts. Despite this inching towards the environmental remediation, MXenes are turning out to be a good choice in need of modifications. MXene surface can adsorb target pollutants via photodegradation, which can be quite complex [93] (Sinopoli et al., 2019). MXenes can adsorb $CO_2$ at elevated temperatures and low partial pressures. Garcia and group put forward the idea of MXene carbides as $CO_2$ absorbers and the effect of thickness in the adsorption. DFT calculations are employed to evaluate the results theoretically and thus opening a gate for experimental endeavors [132] (Morales-García, Mayans-Llorach, Viñes, & Illas, 2019). Adsorption is directly linked with waste water treatment and pollutant management in a very efficient low cost way and the search for a better material to serve the purpose is very important. The paper determines that surface of MXene carbides undergoes exothermic activation of $CO_2$ molecule and can be easily manipulated by changing stoichiometry, composition, and thickness. This paper also points to the structural similarity between transition metal carbides (TMCs) (111) surface and MXene (0001) surface. Now, TMCs are highly unstable and the formation energy of the bulk form is also high. MXenes having the same structural mapping and atomic sites arraying can be considered as a replacement for TMCs. Chen et al. investigated on $CO_2$ capture as well as its possible conversion to products of value like fuel by using MXene-based products. They exclusively focused on $CO_2$ capture, sensing, and conversion of $CO_2$ and also the recent territories that are being explored in this area [133] (Y. Chen et al., 2020). Also, MXene-based composites with suitable functionalization and

introduced to $CO_2$-philic substances can enhance the chance of $CO_2$ capture more. The $CO_2$ adsorption capacity is directly linked to the increased specific surface area and can reach 44.2 mmol $g^{-1}$ [134] (Ding et al., 2018). MXene at this stage is still being explored in terms of synthesizing and stability, so most of the work is focused on energy storage. A lot of experimental work is pending in order to materialize the theoretical predictions being made so far. In a review article regarding the progress in environmental applications of MXene, scientists found that MXene polymer matrices and covalent, non-covalent modifications can provide diverse opportunities of applications. The covalent modifications offer more stable properties and functions when compared to the non-covalent ones. Apart from adsorbing toxic gases, it has been theoretically proved by Zhang et al. that MXene can successfully adsorb and remove uranyl too, which in turn suggests that MXenes can be used in removing radioactive waste [135] (Zexiang He, Huang, Yue, Zhu, & Zhao, 2021). Somehow, the optical and electronic properties of MXene tend to blend, forming a middle ground that actually opens up multiple areas of applications. In biomedical industry, biotoxicity is another hazard to battle and theories of using MXene is being coined provided that the MXene itself does not produce any impurity and maintains stability [136] (J. Chen et al., 2020).

## 6. Future Aspects and Concluding Remarks

Since the discovery of MXenes a decade ago, many breakthroughs have been made in this field and many more are to come in the future undoubtedly. The attention that it gets is well thought of because of the impact it can potentially create in various sectors of commercial application. Despite the extensive study, there are a few lacunas that should be addressed as well as explored more:

- Exploring simpler, feasible, and environment-friendly routes of synthesis in order to take it from small-scale laboratory synthesis to large-scale industrial production on commercial level. It is important that the experimental endeavors have successful industrial outcomes. Especially for a material to have versatile properties, we can have the ambition to make it dominate versatile sectors of applications. For that though, safer and cost effective routes of production is necessary.
- Despite the amount of theoretical studies, their mechanism especially in the field of energy storage is not fully understood. MXenes are being confirmed to have the most impact in the energy sector, for example in supercapacitors, microsupercapacitors, electrodes in batteries, etc. Still, there is clear lacuna in confirming the stability of MXenes when put under these applications. It is yet to be confirmed whether MXenes are capable of producing desirable results on their own or if they need any kind of stable base or matrix to support their longevity. So, more research works are needed to make MXenes really stand out as the best available material for use in energy storage applications at present.
- The stability of MXenes is a real struggle as MXene nanosheets degrade very quickly. The duration of activity of MXenes before they degradehas been pointed out time and again in numerous experimental papers. In order to have them incorporated in devices, stability issues need to be taken care of firsthand.
- Researching more on MXene-based Polymer composites.
- Shifting from titanium carbides to other MXenes classes. The possibilities of new properties and diverse applications cannot be denied of having other MXenes replacing titanium and focusing on nitrogen-based MXenes. There is a clear lack of evidence in order to make a comparison in properties between them. So, this area needs to be explored more.
- Analyzing mixed surface group terminations and their effect on MXene properties. We have seen how surface terminations can affect not only mechanical properties, but also the impact in various experimental works. Terminated MXenes show far better results than bare MXenes and later we found out that the nature of termination is also important. It will be more interesting to find out about mixed surface terminations, which are practically obtained when a MXene is synthesized. The hydrophilicity of

MXenes is the reason of so many properties and potential applications. Mixed surface terminations can make a MXene a multitasking device.

This is an emerging field with loads of possibilities and should be ventured into more. With the vast availability of MAX phases and the possibility of non-MAX phase,MXenes can lead to unique outcomes. A decade of work has given loads of ideas about this class of material for upcoming projects.

**Author Contributions:** Conceptualization, S.B.; Methodology, S.B.; Formal analysis, S.B.; Investigation, S.B.; Resources, S.B.; Data curation, S.B.; Writing—original draft preparation, S.B.; Writing—review, S.B.; Editing, P.S.A.; Visualization, P.S.A.; Supervision, P.S.A. All authors have read and agreed to the published version of the manuscript.

**Funding:** This research received no external funding.

**Institutional Review Board Statement:** Not applicable.

**Informed Consent Statement:** Not applicable.

**Data Availability Statement:** Not applicable.

**Acknowledgments:** It is my proud privilege to express my profound sense of gratitude to Prashant S. Alegaonkar, Department of Physics, Central University of Punjab, Bathinda for his benevolent guidance, which aided a lot and reduced the workload of my review work. His valuable suggestions, constructive counseling, whole hearted encouragement, and critical appreciation through the work and during the preparation of this manuscript has my earnest gratitude. I also extend my thankfulness to the Respected Vice Chancellor, Raghavendra P. Tiwari, Central University of Punjab and Dean, In-charge Academics, Ramakrishna Wusurika for the constant support system and providing a sound working environment for young students like me. I would also like to thank my classmates Akshita Yadav and Amit Saharan and Chandra Shekhar Singh for their support. Discussing with them created a better sense of understanding over the topics, which I felt challenging. I am highly grateful to my parents for their affectionate and moral support. They have always been a source of inspiration for me. Last but not least, I extend my sincere thanks to all those who have helped me in one or the other way during my project work.

**Conflicts of Interest:** The authors declare no conflict of interest.

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
