# Peer review of "MXene: Evolutions in Chemical Synthesis and Recent Advances in Applications"

_surfaces, doi:10.3390/surfaces5010001_

Round 1

Reviewer 1 Report

The revised manuscript looks fine and now can be accepted in this journal. 

Congratulations!

Reviewer 2 Report

The authors improve significantly the manuscript.

Now, it is appropriate for its publication

This manuscript is a resubmission of an earlier submission. The following is a list of the peer review reports and author responses from that submission.

Round 1

Reviewer 1 Report

In this review, the authors, highlighted major breakthroughs of Mxene on synthesis mechanism and characterization. I believe this would be more interesting if the author include the characterization details in Table 1 and potential applications. If possible, please add 1-2 figures on the application section.

Thanks

Reviewer 2 Report

This is a review paper on the subject of “Mxene” materials which means that gathering comprehensive information and literature of the topic and then comparing the results and interpreting them as well as comprehensive discussion. I have not found this manuscript in this way and what I found is non-consistence information from different papers that just re-written as a collection of information. Review paper must debate on the different aspects of the issue, combine and compare the results in different aspects and points of view and plot them. While here, only the pics and schematics from others were shown again. The advantages and disadvantages of these new gage 2D materials and their comparison should be listed in the review paper.

A further complication of the lack of scientific understanding is the problem in the organization and logic of this review manuscript. Instead of dealing with important points individually and specifically, the authors mixed many things together throughout the paper and, therefore, failed to provide a clear and convincing discussion. Therefore, this manuscript cannot be accepted in the “Surfaces” journal in the present form. There are also numerous mistakes or meaningless, vague statements in the paper. Some examples are given below:

  1. The title of the manuscript must be improved to highlight the overall subject of this study. It is not acceptable in the present form. Please pay attention that, this is not a book chapter or not an individual technical report.

  1. It is beneficial to provide a separate timeline-based figure of the evolution of Mxene and important breakthroughs. But as can be seen, Fig. 3 is not a figure at all. Providing step by step or follow up flow chart or something like this can improve the quality and readability of this figure.

  1. The main issue with the paper in its present form is the introductory part, which needs to be expanded. The overview section is not acceptable and should be a separate chapter. The introduction should be substantially expanded. This is particularly important for a comprehensive review, which should have an introduction aimed at a broad audience, providing an accessible introduction to the topic for a general reader (especially focused on the family of 2D material-Mxene).

  1. The “Results” section must be substantially improved for increasing the quality of the review paper. Authors are encouraged to expand the main characteristics which have been collected as the results in this section.

  1. Furthermore, the review should provide an intellectual contribution/insight/outlook beyond what is already covered in existing reviews on the same or closely related topics. It should be explicitly clear in the paper (usually but not always in the introduction) how this is achieved. For example, the review should provide a distinct contribution from reviews on Mxene material, as well as its general properties, applications, characterization, and new findings in this field. Making the distinction clear is easy, but just remember that you are writing also with non-specialist readers and students new to the topic in mind – be explicit and clear on the difference.

  1. Page 4 and 5: there is a detailed table (Table 1: Summary of Mxenes prepared by HF treatment) in the paper. Carefully vetted table(s) of essential data from the literature can be included, but please take care to avoid excessive detail of little interest to the readers. A review should summarize the essence of all these details in an overview format accessible to both generalist and specialist readers. Also, provide an individual column for Ref. inside of the table.

  1. This review manuscript should provide a critical assessment of the topic and the literature and be carefully fact-checked. A literature survey on the subject of Mxenes and their properties alone is not sufficient.

  1. Also, it is important that the article is written in a pedagogical and logical manner with a clear connection between sections and different parts of the paper. As can be seen, the submitted manuscript lacks a clear division between the sections.

  1. In the introduction, you write “Good flexibility owing to their 2D morphology and sheet-like structure is the particular feature of Mxenes that makes them outstanding for composite formation.” Please elaborate on how. It is imperative that the review provide an intellectual contribution/insight/outlook beyond what is already covered in existing reviews, which should be explicitly cited here.

  1. The review paper should explain the mechanisms and phenomena on the synthesis, formation, and properties of Mxenes not just report the results. In most of the cases, I found just a report without any explanation. Therefore, the existed polymerization, delamination mechanisms together with the reaction mechanism(s) of the hydrothermal etching route must be expanded in this manuscript.

  1. The English level of the manuscript should be improved there are some grammatical and dictation mistakes in the manuscript.

  2. The references format must be double-checked by the authors. New references also must be used for increasing the quality and readability of this paper.
  3. The overall number of figures and tables is not sufficient for this review manuscript. 

Reviewer 3 Report

In this review, the authors revise the state-of-the-art of MXenes. There is a large description of the synthesis and characterization. Some properties are discussed those related with mechanical, electronic, and optical issues.

In my opinion, this review needs to be improved significantly. Especially in the section of properties. I think that the paper will improve notoriously discussing the catalytic and photocatalytic properties of MXenes. Recently MXenes has been proposed to capture CO2, nitrogen fixation, water dissociation. Theoretical and experimental evidence must be discussed. On the other hand, heterostructures based on MXenes are suitable charge separators to prolong the life of exciton. So, promising for photocatalysis.

These aspects constitute the last applications of MXenes and they are mandatory to include in this work.

In addition, the section Future aspects and concluding remarks has to be extended by discussing each one of the points mentioned on it.